# Meiotic sex in Chagas disease parasite *Trypanosoma cruzi*

Philipp Schwabl [1], Hideo Imamura [2], Frederik Van den Broeck [2], Jaime A. Costales [3], Jalil Maiguashca-Sánchez [3], Michael A. Miles[4], Bjorn Andersson [5], Mario J. Grijalva [3,6,7] & Martin S. Llewellyn [1,7]

Genetic exchange enables parasites to rapidly transform disease phenotypes and exploit new host populations. *Trypanosoma cruzi*, the parasitic agent of Chagas disease and a public health concern throughout Latin America, has for decades been presumed to exchange genetic material rarely and without classic meiotic sex. We present compelling evidence from 45 genomes sequenced from southern Ecuador that *T. cruzi* in fact maintains truly sexual, panmictic groups that can occur alongside others that remain highly clonal after past hybridization events. These groups with divergent reproductive strategies appear genetically isolated despite possible co-occurrence in vectors and hosts. We propose biological explanations for the fine-scale disconnectivity we observe and discuss the epidemiological consequences of flexible reproductive modes. Our study reinvigorates the hunt for the site of genetic exchange in the *T. cruzi* life cycle, provides tools to define the genetic determinants of parasite virulence, and reforms longstanding theory on clonality in trypanosomatid parasites.

[1] Institute of Biodiversity, Animal Health & Comparative Medicine, University of Glasgow, Glasgow G12 8QQ, UK. [2] Unit of Molecular Parasitology, Institute of Tropical Medicine Antwerp, 155 Nationalestraat, 2000 Antwerp, Belgium. [3] Center for Research on Health in Latin America, School of Biological Sciences, Pontifical Catholic University of Ecuador, Quito, Ecuador. [4] London School of Hygiene & Tropical Medicine, Keppel Street, London WC1E 7HT, UK. [5] Department of Cell and Molecular Biology, Science for Life Laboratory, Karolinska Institutet, Biomedicum 9C, 171 77 Stockholm, Sweden. [6] Infectious and Tropical Disease Institute, Biomedical Sciences Department, Heritage College of Osteopathic Medicine, Ohio University, 45701 Athens, OH, USA. [7] These authors contributed equally: Mario J. Grijalva, Martin S. Llewellyn. Correspondence and requests for materials should be addressed to M.S.L. (email: martin. llewellyn@glasgow.ac.uk)

*T*rypanosoma cruzi is a kinetoplastid parasite and the causative agent of Chagas disease in Latin America, where ca. six million people are currently infected[1]. Mucosal or abrasion contact with the infected feces of hematophagous triatomines constitutes the primary mode of *T. cruzi* transmission. Infection with *T. cruzi* results in chronic Chagas disease in 30–40% of cases, characterized by a spectrum of fatal cardiac and intestinal pathologies. Early-stage acute Chagas disease can also be fatal, especially among infants and in orally transmitted outbreaks of the disease[2]. *T. cruzi* transmission is a zoonosis maintained by numerous species of triatomine insects and hundreds of different species of mammals[3].

The Trypanosomatidae, the family to which *T. cruzi* belongs, is a monophyletic group of obligate parasites and includes several species of medical and veterinary importance – e.g., *Trypanosoma brucei* ssp., *Leishmania* spp., *Trypanosoma vivax*, and *Trypanosoma congolense*[4]. The Trypanosomatidae are early branching eukaryotes in evolutionary terms and share many biological characteristics, including the process of U-indel RNA editing in the kinetoplast[5] and polycistronic transcription control[6]. Despite their basal status, the Trypanosomatidae possess much of the core meiotic machinery of higher eukaryotes[7]. However, the extent to which such machinery might actually support genetic exchange within trypanosomatid species has been slow to come to light[8]. Establishing the occurrence of regular meiotic recombination in *T. b. brucei* has taken decades of laboratory and field research; not until 2014 was haploid gamete production (coincident to peak meiosis-specific gene expression) confirmed by fluorescence microscopy as a normal phase of development in the vector's salivary gland[9–11]. More recently, genome-scale signatures of meiosis have also been detected in *T. congolense*[12,13]. In contrast, robust genomic evidence now suggests that the human-infective *T. b. gambiense* subspecies is completely asexual[14]. Life histories in *Leishmania* seem no less complex. Despite a clear propensity for mitotic clonality, sporadic sexual hybrid formation appears to underlie important diversification events both within and between species[15,16], and meiotic offspring are readily produced in laboratory crosses[17,18]. An alternation of clonal and sexual, endogamic reproduction has also been proposed to define population genetic structure in the *Viannia* complex[19].

*T. cruzi* is the last of the Tritryps (*Leishmania* spp., *T. brucei* ssp. and *T. cruzi*) for which the extent and mechanism of genetic exchange remains to be fully elucidated. Limited evidence for genetic recombination has been observed in the field[20,21] although inappropriate study designs, genetic marker systems of insufficient resolution, and low genetic diversity in study populations have all hampered interpretation of the data[8]. Furthermore, the parasexual mechanism of genetic exchange proposed for *T. cruzi* based on a single experimental cross – one of whole-genome fusion followed by stochastic chromosomal decay and return to diploidy[22] – has been irreconcilable with patterns of somy and genetic diversity observed in natural populations[20,23,24]. This lack of clarity has lead some to propose *T. cruzi* as a paradigm for Predominant Clonal Evolution (PCE)[25,26] in parasitic protozoa – an idea which may not reflect biological reality.

To address this fundamental knowledge gap in the biology of trypanosomatids, in this study we generate whole-genome sequence data from 45 *T. cruzi* Discrete Typing Unit I clones, as well as several non-cloned *T. cruzi* strains, collected from triatomine vectors and mammalian hosts in an endemic transmission focus in Loja Province, southern Ecuador. After mapping sequences against a recent PacBio sequence assembly[27], we explore patterns of population structure and genetic recombination. Our data reveal that *T. cruzi* does indeed reproduce sexually at high frequency via a mechanism consistent with classic meiosis. However, we demonstrate that parasite groups with radically distinct reproductive modes also co-occur at the same transmission focus. As the last medically important trypanosome for which meiosis has not yet been demonstrated in lab or field, our data on *T. cruzi* make a significant contribution towards the consolidation of current theories around genetic exchange in the Trypanosomatidae.

## Results

**Extensive genetic divergence between sympatric parasites.** Paired-end sequence reads from 45 single-clone and 14 non-cloned *T. cruzi* cultures aligned to the reference assembly (*T. cruzi* TcI X10/1 Sylvio) at a mean depth of 27× , ranging between 13 and 64× (Supplementary Table 1). The *T. cruzi* genome is highly repetitive, especially in the sub-telomeric regions[27]. Extensive optimization of variant filtration and masking was therefore undertaken before a total of 206,619 SNP sites could be robustly identified against the reference (see *Methods*). Including only single-clone *T. cruzi* cultures founded from individual parasites in the laboratory, 130,996 SNP sites were identified that clearly separated our samples into two highly distinct phylogenetic clusters within the small study area (Fig. 1, Supplementary Fig. 1, Supplementary Table 1). Cluster 1 contained 15 of 17 clones isolated from triatomine vectors and mammal hosts captured in the community of Bella Maria. Cluster 2 contained 2 clones from Bella Maria, 11 clones from nearby Ardanza (ca. 7 km south), as well as 3 clones from Gerinoma and 12 from El Huayco study sites ca. 35 km northwest of Bella Maria. Two clones from Santa Rita (near El Huayco) associated to Cluster 1. Unsupervised k-means clustering further confirmed two major clusters (i.e., $k = 2$) among the samples, although mild improvements to model fit continued through to $k = 6$ (Supplementary Fig. 2).

To further detail parasite population genetic substructure within and across potentially multiclonal infections (multiple clones were often sampled from a single vector/host individual – see clone ID prefixes), we reconstructed each phased genome as a mosaic of haplo-segments sharing ancestry with other samples of the dataset[28]. In the resultant co-ancestry matrix (Fig. 2), which also includes isolates that had not been subject to solid-phase cloning, intensity of haplotype-sharing (see color scale) increased within both clusters relative to the spatial origin of each clone, with the exception of TCQ_3087 (sampled in Bella Maria but associated to Cluster 2) and TRT_3949 clones (sampled near El Huayco but associated to Cluster 1). Importantly, four non-cloned samples (TBM_3131_MIX, TBR_4307_MIX and TRT_4082_MIX, cultured from the triatomine species *Rhodnius ecuadoriensis*, and MBC_1529_MIX, cultured from the rodent species *Simosciurus nebouxii*) showed shared ancestry across Clusters 1 and 2. Clones derived from the same strains did not show shared ancestry. These data may indicate the presence of multiclonal infections in which parasites from these distinct groups co-occur in the same vectors and hosts (Supplementary Table 1).

**Sympatric Mendelian and non-Mendelian genetic traits.** To explore eco-evolutionary processes potentially underpinning sympatric divergence in *T. cruzi*, we established key metrics of population genetic structure at different sites. Among the 15 Bella Maria clones of Cluster 1, allele frequencies at variable loci matched those predicted for random mating, with estimated inbreeding coefficients predominantly near zero ($\bar{x} = -0.11$, $\sigma = 0.38$; Supplementary Fig. 3) and 87,600 of 96,691 (91%) variant loci meeting expectations for Hardy–Weinberg equilibrium (Table 1). Heterozygosity was unevenly distributed across each chromosome

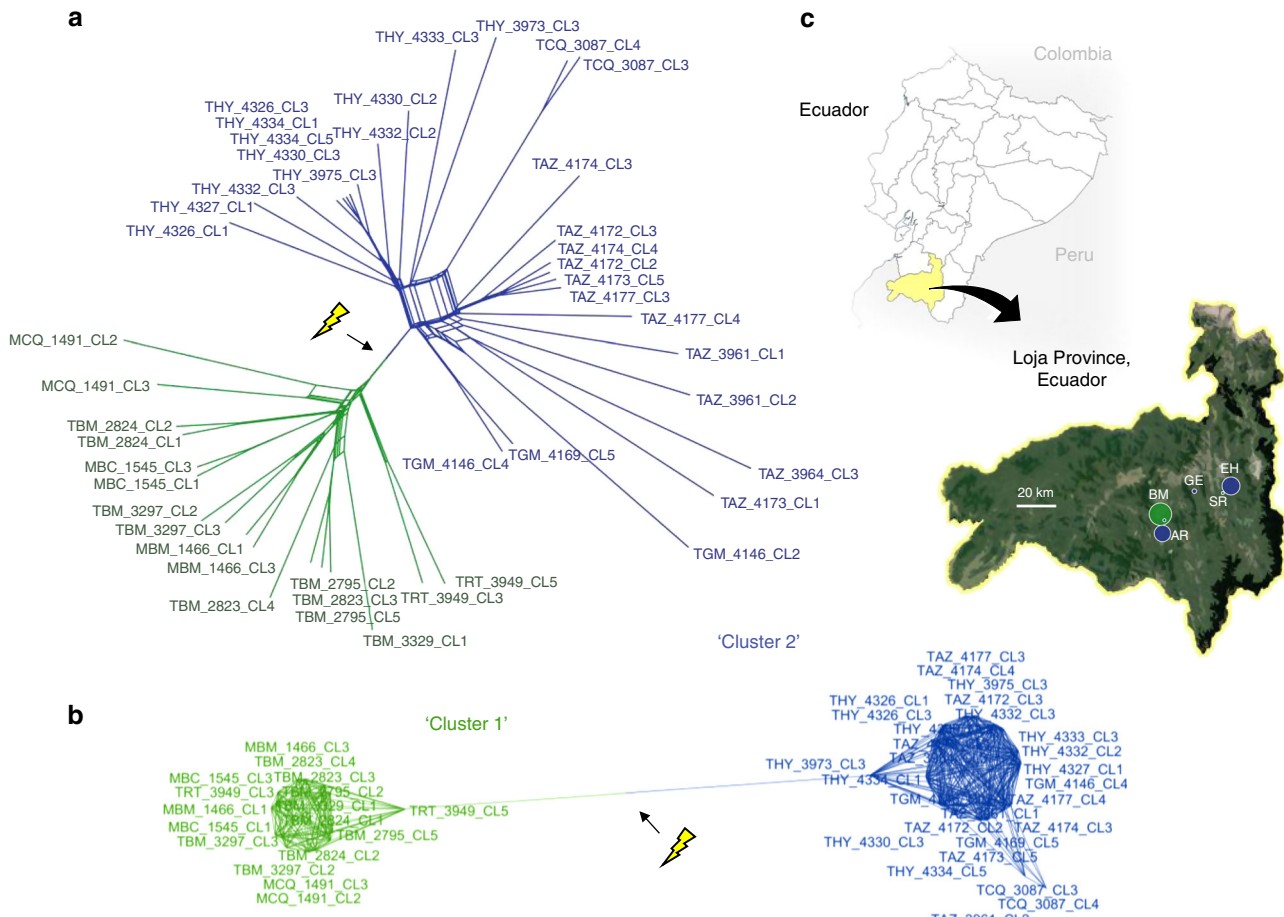

**Fig. 1** Phylogenomic relationships among *T. cruzi* I clones from southern Ecuador. **a** Data are represented as a split network by the Neighbor-Net algorithm[68]. Pairwise genetic distances are defined as the proportion of non-shared genotypes across all biallelic SNP sites for which genotypes are called in >40 individuals ($n = 68,449$). Arrow (and flash) indicate a strong, unambiguous break in gene-flow between two reticulate assemblages, Cluster 1 (green) and Cluster 2 (blue). Though non-treelike phylogenetic models are better suited to the data, a maximum-likelihood tree is also provided for comparison in Supplementary Fig. 1. **b** A minimum-spanning network[80] further illustrates the genetic disconnectivity between Clusters 1 and 2. Multi-furcating nodes are arranged such that cumulative edge distance is minimized among samples. Pairwise genetic distances are haplotype-based, defined as the proportion of non-shared alleles across all SNP sites for which genotypes are called for all individuals ($n = 7,392$). **c** Sampling regions in Loja Province, Ecuador, are abbreviated as BM Bella Maria, AR Ardanza, EH El Huayco, SR Santa Rita, and GE Gerinoma. Point sizes correspond to sample sizes and colors correspond to cluster membership (see phylogenetic networks). The upper map is adapted from https://d-maps.com/carte.php?num_car=3400&lang=en (d-maps.com). The close-up uses Landsat imagery courtesy of the U.S. Geological Survey

(see below), fixed at only 4% (2134/58,102) polymorphic sites (Table 1) and often interrupted by long runs (>100 kb) of homozygosity (Supplementary Table 2). Patterns of allelic diversity in Cluster 2 groups were highly distinct to those observed in Cluster 1. In El Huayco and Ardanza, departures from Hardy–Weinberg equilibrium were noted at 42% and 46% of total polymorphic sites (Table 1). High levels of heterozygosity (Supplementary Fig. 3) extended continuously across all chromosomes (see below). Seventy-six per cent (44,945/58,980) of heterozygous loci occurred as fixed SNPs within El Huayco and 78% (45,287/58,392) occurred as such in Ardanza. Unlike in Bella Maria, long runs of homozygosity occurred in just two of 23 samples (1 instance each) in El Huayco and Ardanza (Supplementary Table 2). Analysis repeated with only one random clone per vector/host showed the same strong contrasts between Clusters 1 and 2, but low sample sizes restricted significance tests (Supplementary Table 3, Supplementary Fig. 4).

As well as extreme differences in the frequency and genomic distribution of heterozygous sites, other features of allelic diversity also diverged starkly among our sympatric study groups. Sliding window analyses of haplotype-sharing among individuals

revealed, on average, much larger contiguous blocks of shared identity among samples from El Huayco and Ardanza (Cluster 2) than Bella Maria (Cluster 1) (Supplementary Fig. 5) despite lower nucleotide diversity ($\pi$) in the latter group (Table 1). Short blocks of shared identity among samples could be consistent with meiotic recombination in Bella Maria and we undertook further analyses to establish if this was the case.

**Linkage decay and rates of meiotic recombination.** In sexually recombining organisms, pairwise SNP-associations ($r^2$) are predicted to decay with map distance due to crossover that occurs between homologous chromosomes during meiosis. We plotted $r^2$ against pairwise map distance for all diagnostic SNP loci identified at Bella Maria. Figure 3a depicts results for chromosome 1, with linkage declining sharply in the first few kilobases, then more gradually and approaching zero near 60 kb. Linkage decay was apparent on other chromosomes examined (chromosomes 5, 21, and 26 (Fig. 3b)). These chromosomes were selected based on their superior mapping quality, avoiding those with extensive masking (Supplementary Fig. 6). Decay curves were

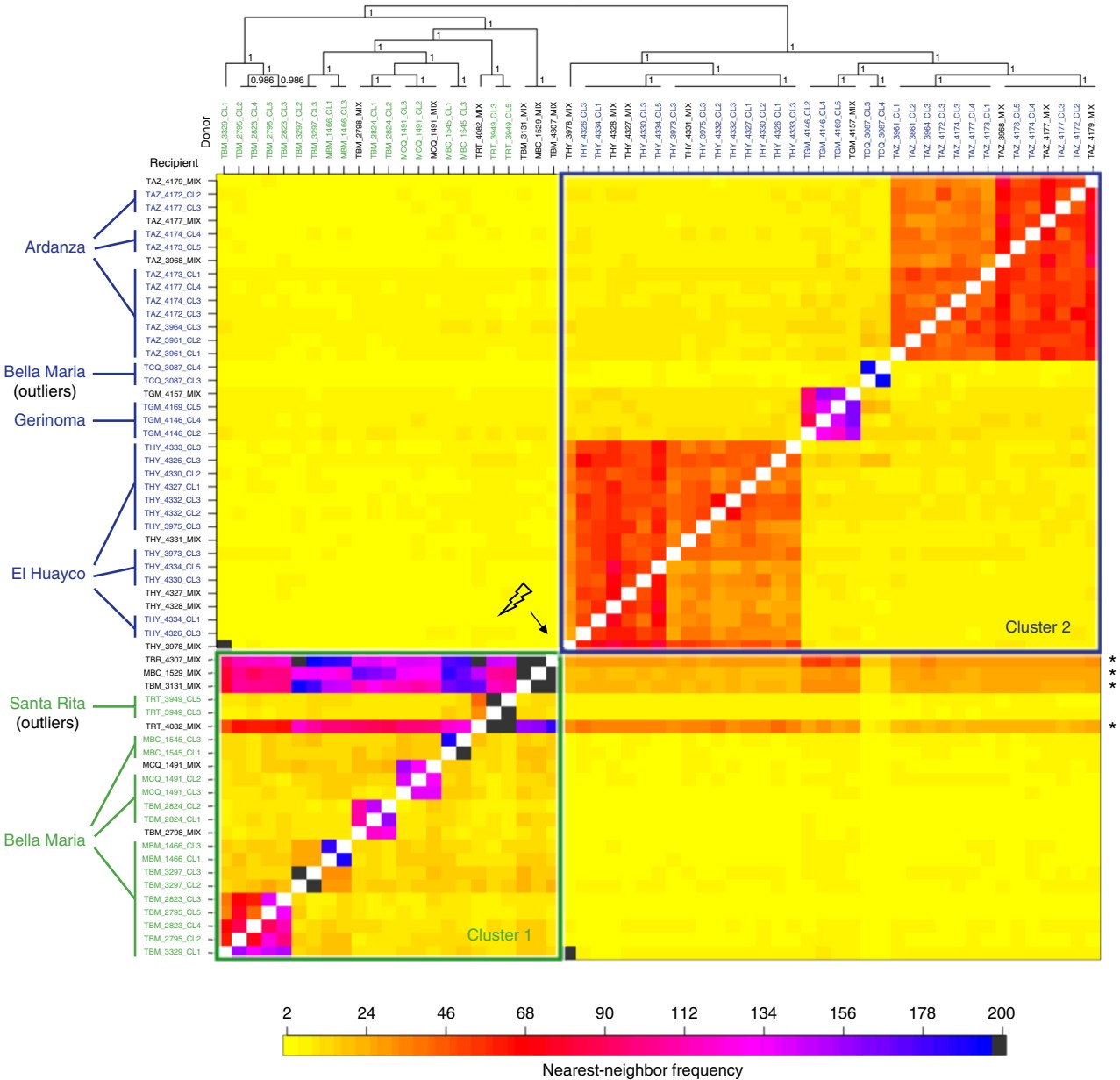

**Fig. 2** Haplotype co-ancestry among *T. cruzi* I clones from southern Ecuador. The heatmap of co-ancestry is based on a sorted haplotype co-ancestry matrix $x_{ij}$, which estimates the number of discrete segments of genome *i* that are most closely related to the corresponding segment of genome *j*. These nearest-neighbor relationships from fineSTRUCTURE[28] analysis are sorted such that samples clustered along the diagonal are those that most share recent genealogical events, and pairwise comparisons outside of the diagonal indicate levels of genetic connectivity among these clusters. The matrix also includes 'genomes' of non-cloned *T. cruzi* cultures. Strong horizontal banding points to the accumulation of diversity from throughout the dataset in four of these original infections. Cell color represents the frequency of nearest-neighbor relationships for each sample pair, increasing from yellow (2) through red (68) and pink (134) to black (200). Four anomalous (outlier) samples are described further on in main text. Analysis uses 110,326 phased SNP sites

**Table 1 Population genetic descriptive metrics for *T. cruzi* I clones from Bella Maria (Cluster 1), El Huayco and Ardanza (Cluster 2)**

| Group (n) | PS | Median $\pi$ | Median $\theta$ | PS at MAF > 0.05 | PRS (vs. BM/EH/ AR) | SS | PS in HW Equilibrium | HS | Fixed HS |
|---|---|---|---|---|---|---|---|---|---|
| Bella Maria (15) | 96,691 | 0.09 | 0.001 | 48% | 0/40,177/40,262 | 14,013 | 87,500 | 58,102 | 2134 |
| El Huayco (12) | 80,052 | 0.15 | 0.001 | 70% | 23,538/0/18,016 | 4525 | 33,980 | 58,980 | 44,945 |
| Ardanza (11) | 78,325 | 0.16 | 0.001 | 71% | 21,896/16,289/0 | 6064 | 35,799 | 58,392 | 45,287 |

Please see Supplementary Table 3 for analogous results from analysis repeated with only one parasite clone per vector/host. *PS* polymorphic sites, *π* nucleotide diversity, per site, *θ* Watterson estimator, per site, *MAF* within-group minor allele frequency, *PRS* private sites, *SS* singleton sites, *HW* Hardy–Weinberg, *HS* heterozygous sites

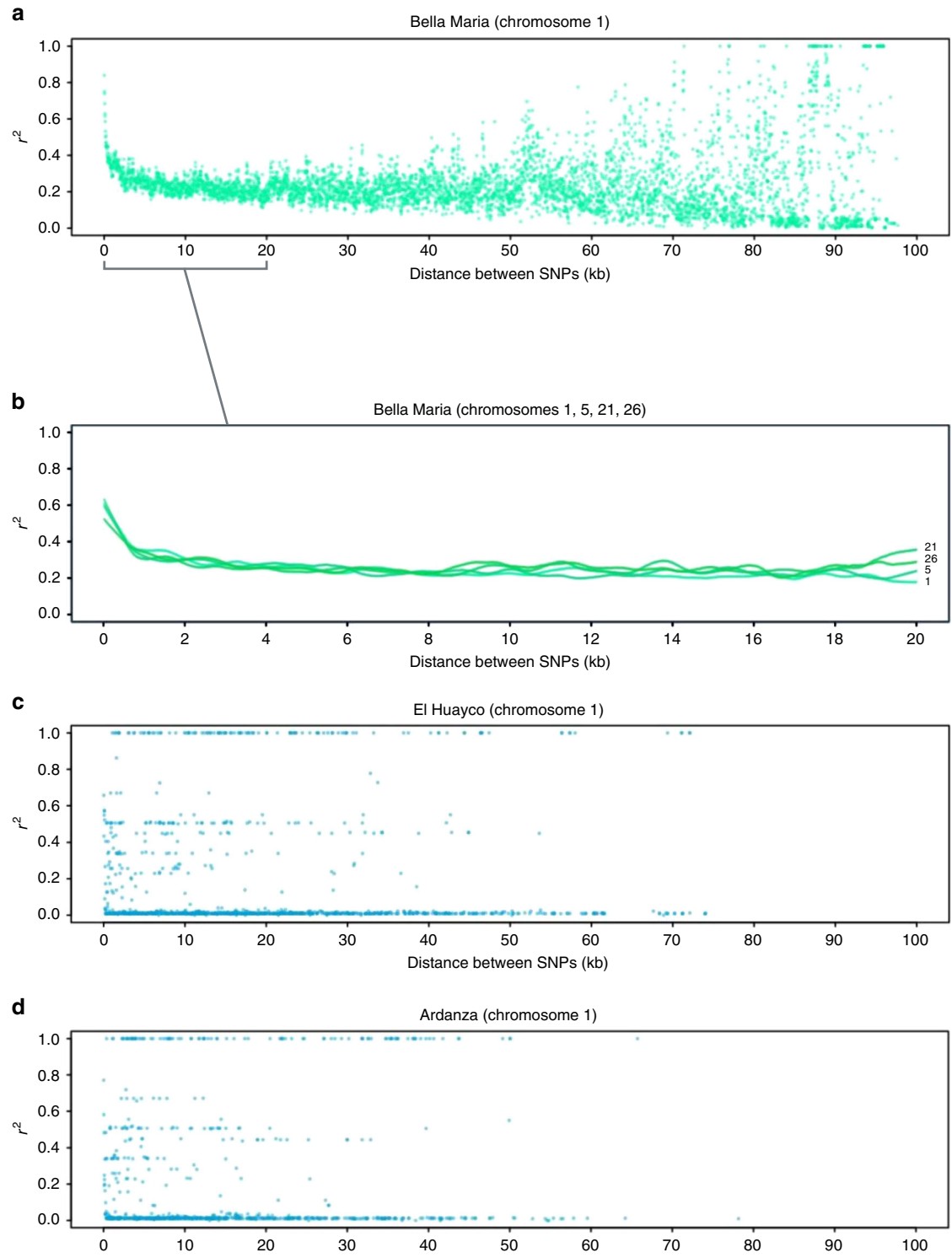

**Fig. 3** Linkage decay and different rates of recombination in *T. cruzi* I groups. **a** Decay of linkage disequilibrium on chromosome 1 for *T. cruzi* I clones from Bella Maria. Average pairwise linkage values ($r^2$) among SNP sites present in at least 90% individuals ($n = 5373$) are plotted for map distance classes between 0 and 100 kb. **b** Local regression curves for the decay of linkage disequilibrium on chromosomes 1, 5, 21, and 26 for *T. cruzi* I clones from Bella Maria. **c, d** Lack of linkage decay on chromosome 1 for *T. cruzi* I clones from El Huayco (4093 SNPs) and Ardanza (3306 SNPs). Average pairwise linkage values ($r^2$) are plotted against distance classes as for Bella Maria above

robust to reduction of the dataset to include only one clone per infection (Supplementary Fig. 7a) and also emerged in analysis restricted to core sequence regions (genes syntenous to *T. b. brucei* and *L. major*) (Supplementary Fig. 7b). In contrast to clones from Bella Maria, analyses of linkage decay for clones from El Huayco and Ardanza showed no relationship between $r^2$ and map distance. Rather, complete and intermediate linkage, as well as an abundance of random variant-associations, featured continuously through all distance classes on the same chromosomes surveyed in Bella Maria – e.g., chromosome 1 (Fig. 3c, d).

**Table 2 Composite-likelihood approximation of the population recombination parameter ρ**

| Region | Group (n) | Median ρ (Morgans kb⁻¹) | 95% Confidence Interval |
|---|---|---|---|
| Chr. 1 | Bella Maria (15) | 0.424 | 0.370–0.562 |
| Chr. 5 | Bella Maria (15) | 0.549 | 0.400–0.647 |
| Chr. 21 | Bella Maria (15) | 0.534 | 0.514–0.560 |
| Chr. 26 | Bella Maria (15) | 0.357 | 0.338–0.392 |
| Chr. 1 | El Huayco (12) | 0.004 | 0.004–0.004 |
| Chr. 5 | El Huayco (12) | 0.002 | 0.001–0.004 |
| Chr. 21 | El Huayco (12) | 0.002 | 0.001–0.003 |
| Chr. 26 | El Huayco (12) | 0.005 | 0.002–0.016 |
| Chr. 1 | Ardanza (11) | 0.005 | 0.005–0.005 |
| Chr. 5 | Ardanza (11) | 0.003 | 0.002–0.003 |
| Chr. 21 | Ardanza (11) | 0.002 | 0.000–0.004 |
| Chr. 26 | Ardanza (11) | 0.002 | 0.001–0.002 |
| Chr. 1, simulated | FSC_r (10) | 78.886 | 77.023–80.739 |
| Chr. 1, simulated | FSC_n (10) | 0.001 | 0.000–0.007 |
| Chr. 1, simulated | BS_n (10) | 0.000 | 0.000–0.000 |

Positive approximations of ρ for *T. cruzi* I isolates from Bella Maria differ from estimates derived for synthetic non-recombinant controls. The FSC_n control represents ten 3.1 Mb chromosomes simulated without recombination in fastsimcoal2[77]. The confidence interval around ρ estimates for FSC_n overlaps zero. It also overlaps estimates for El Huayco, Ardanza and BS_n, a second synthetic non-recombinant dataset generated by BAMSurgeon[76] simulation approach (see *Methods*). Results from chromosome simulation with the recombination rate *r* set to 3.2 × 10⁻⁴ (FSC_r) demonstrate the sensitivity of the LDhat[74] interval program applied to 100,000 diploid individuals under a finite-sites model of evolution

We estimated the frequency of meiosis ($N_\rho/N_\theta$) in our study groups by comparing two different estimates of effective population size. The first estimate, $N_\rho$, is based on recombinational diversity observed in the sample and represents the number of cells derived from mating. The second, $N_\theta$, is based on mutational diversity and represents the total number of cells, irrespective of sexual or mitotic origin (see *Methods*). As in linkage decay analysis, we considered the best-mapping chromosomes 1, 5, 21, and 26. Values of ρ for Bella Maria suggested ca. 3 meioses per 1000 mitotic events in this group. In contrast, all approximations of ρ for El Huayco and Ardanza fell within confidence limits of the simulated, non-recombinant FSC_n control. These limits also contained ρ = 0 (Table 2).

The intra-chromosomal recombination detected for Bella Maria was further explored by aligning individual windowed alternate allele frequency means (AAFM) among clones (Fig. 4a, b). As indicated previously (Supplementary Table 2), sample genomes in Bella Maria presented intermittent patches of high homozygosity (where AAFM approaches 1), and these patches were often shared by variable subsets of clones (see windows with red fill color in Fig. 4a, b). Given that SNP was predominantly biallelic (<1.5% sites with >2 alleles) in Bella Maria as well as in Cluster 2, these patches corresponded directly to abrupt segmental increases in sequence similarity between clones (see SNP alignment in Supplementary Fig. 8, expanded in Supplementary Figs. 9 (chr. 1) and 10 (genome-wide)). Mosaic patterns of recombination between Bella Maria clones were confirmed by fluctuating intra-chromosomal genealogies established using sliding-window neighbor-joining topology weighting in Twisst[29]. Figure 4c shows how strong support for various different tree topologies emerges sporadically throughout chromosome 1. Such mosaicism occurred genome-wide for most samples from Bella Maria (Supplementary Figs. 10 and 11b), but very infrequently in Cluster 2 (Fig. 4d, Supplementary Figs. 10 and 11).

**Evidence of independent chromosomal ancestries in all groups.** Apart from disrupting sequence patterns within chromosomes, sexual reproduction breaks up associations between chromosomes within the genome. Given sufficient population diversity, therefore, incongruent phylogenies are expected depending on the chromosome used to construct them. As one might expect given estimated rates of meiotic sex in this group, we encountered many such incongruences among Bella Maria clones belonging to Cluster 1 (Fig. 5). Intriguingly, ancestries among several clones from Cluster 2 also showed signs of incongruence at the chromosomal level (Fig. 5). We also recognized these varying affinities among El Huayco, Ardanza, and Gerinoma clones in discriminant analysis results for higher k-means solutions (e.g., see individual membership probabilities for $k = 5$ in Supplementary Fig. 2b) and noted occasional shifts to common homozygosity unrelated to coding vs. non-coding sequence annotation in painted genomes (e.g., see chromosomes 6, 14, and 41 in Supplementary Fig. 10). While subtle, such segmental changes argued against divergence in strict isolation among Cluster 2 clones: if not classic chromosomal reassortment, some form of introgression appears to have occurred in this group.

**Signatures of hybridization in highly heterozygous genomes.** Of 80,052 SNP sites that differed from the TcI-Sylvio reference genome in El Huayco, 62,036 also differed in Ardanza, and >50% of this polymorphism occurred as fixed heterozygous loci across the two groups. These observations, supported by population genetic statistics (see Table 1) and phylogenetic similarity (Figs. 1 and 2), provided indications of potential shared ancestry across clones of Cluster 2, and possibly a hybrid origin of this group.

To further explore potential hybrid origins of Cluster 2 clones, we first expanded our previous within-group windowed haplotype analyses to include comparisons of *T. cruzi* clones between El Huayco and Ardanza groups (Supplementary Fig. 12a). These between-group pairwise comparisons of phased SNPs exposed the frequent co-occurrence of haplotype polymorphism in clones from Ardanza and El Huayco (but not clones from Bella Maria). Reaching up to 180 kb, shared haplotype segments appeared similar in size to those found in pairwise comparisons within Bella Maria (Supplementary Fig. 12b) and suggest recent genetic connectivity throughout Cluster 2. This between-group connectivity is also apparent upon careful examination of the co-ancestry matrix in Fig. 2.

Patches of low differentiation observed in pairwise comparisons within El Huayco (Supplementary Fig. 12c) and within Ardanza (Supplementary Fig. 12d) often involved both haplotypes. Unlike cases of haplotype-sharing described above, these were stretches of similar or identical heterozygous diplotypes (i.e., phased regions in which haplotype A occurs in samples 1 and 2, and also haplotype B occurs in samples 1 and 2) interrupting otherwise dissimilar heterozygous sequence between two clones. Such diplotype sharing within groups of Cluster 2 extended far beyond 180 kb, often to the end of the chromosome. The same phenomenon was rarely observed in comparisons of clones within Bella Maria (Supplementary Fig. 12f) and could point to the passage of Cluster 2 clones through an ancestral polyploid state.

To characterize ploidy variation in Cluster 2, somy analysis was undertaken (Fig. 6). Chromosome-wide deviations in variant allele fraction and total read-depth suggested full-chromosome trisomies in ten samples (Fig. 6a, b), with highest rates in THY_3975, THY_4326 and THY_4332 clones (>10 trisomies each). Of 21 chromosomes with apparent trisomy, ten appeared trisomic in ≥5 samples, with similar biases apparent in El Huayco and Ardanza (e.g., chromosomes 19, 25, and 39). To explore the intra-clonal stability of somy variation over time, we re-sequenced three aneuploid clones after sample cryo-preservation and re-expansion in liquid culture (results are denoted with T2 suffix). While inferred karyotypes of THY_4326_CL1_T2 and TAZ_4174_CL4_T2 matched initial

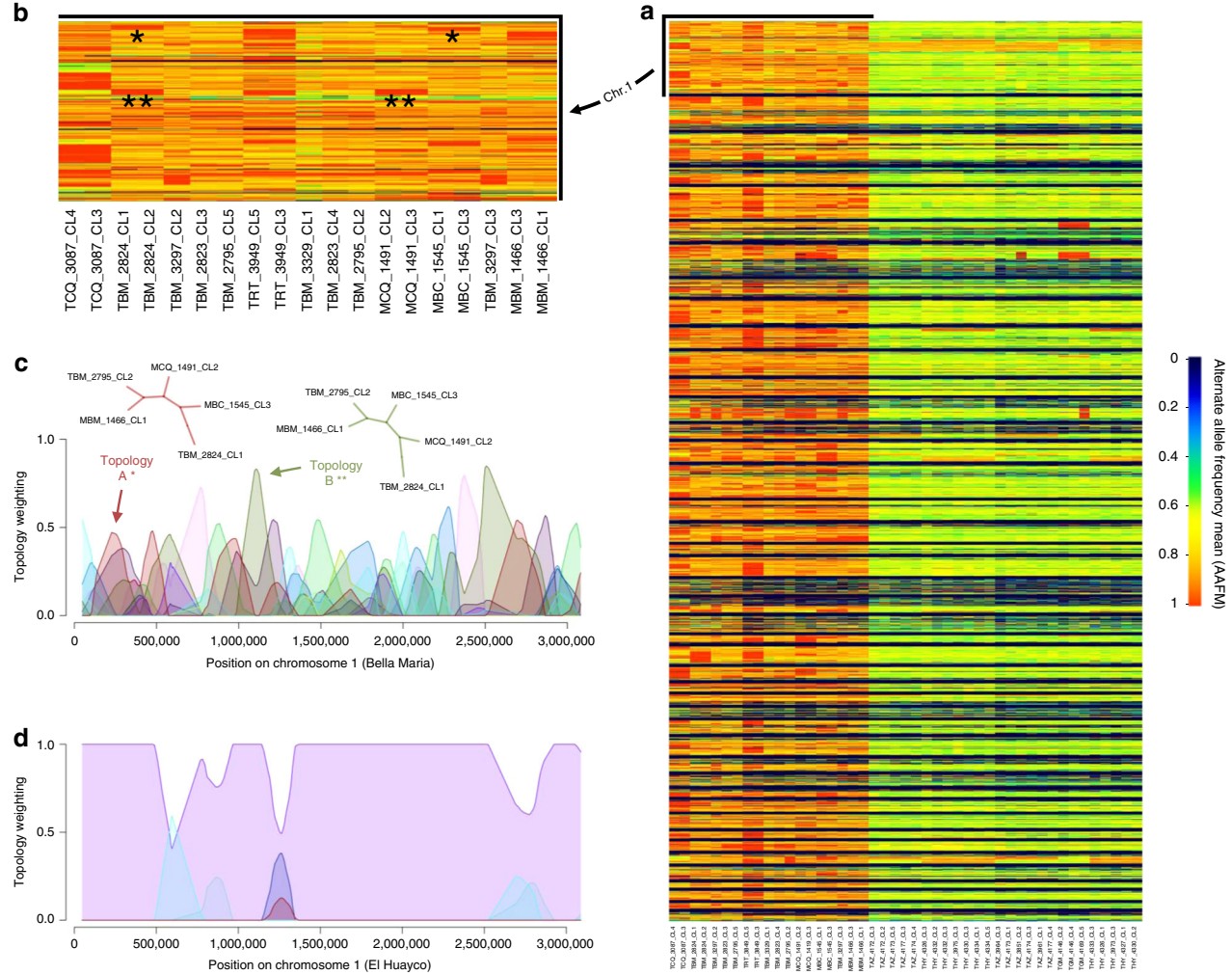

**Fig. 4** Genome-wide heterozygosity patterns and intra-chromosomal mosaics in *T. cruzi* I clones. In **a**, each column represents the genome of one clone, considering the dataset's total 130,996 SNPs. Rows within each column represent consecutive 5 kb sequence bins. Alternate allele frequency means (AAFM) determine the color of each bin – blue (0) through green (0.5) to red (1). Clones from Bella Maria tend to carry patchy homozygosity while those of Cluster 2 appear highly heterozygous throughout the genome. Isolated tracts of high homozygosity (i.e., red patches) shared between pairs of Bella Maria clones imply sudden sequence similarity and fluctuating phylogenetic relationships inconsistent with divergence through drift. **b** provides a close-up on chromosome 1. **c**, **d** demonstrate the impact of this intra-chromosomal mosaicism on the topology of phylogenetic trees derived in a sliding window across chromosome 1. Multiple incongruent topologies are present in Bella Maria (**c**), consistent with widespread genetic exchange. Only a single topology dominates for samples of El Huayco **d**, consistent with limited genetic exchange in Cluster 2. An example of how AAFM heatmaps correspond to topology analyses is indicated in the heatmap close-up **b** and tree topologies in **c**: a shared red patch between TBM_2824_CL1 and MBC_1545_CL3 corresponds to neighbor-joining tree topology A in **c**. Later, near ca. 1100 kb, a shared patch of high AAFM between TBM_2824_CL1 and MCQ_1491_CL2 begins. This patch occurs where tree topology B best describes phylogenetic relationships in Bella Maria. Topology B is identical to topology A except for the replacement of MBC_1545_CL3 by MCQ_1491_CL2 as nearest neighbor to TBM_2824_CL1

results (Supplementary Fig. 13a, b), several aneuploid chromosomes in THY_4332_CL3 appeared to have reverted to the disomic state by time T2 (Supplementary Fig. 13c). We also examined ploidy in subclones of each re-passaged clone. No significant variation occurred among the three subclones obtained from THY_4332_CL3_T2 nor between the two obtained from THY_4326_CL1_T2, each with a karyotype matching that of the parental clone (Supplementary Fig. 13b, c). Somy estimates for the single subclone obtained from TAZ_4174_CL4_T2, however, were inconsistent to the progenitor karyotype (Supplementary Fig. 13a).

In contrast to karyotypic variation in Cluster 2, we found minimal rates of aneuploidy in Cluster 1 (Bella Maria). With the exception of TBM_2824 clones (trisomic for chromosomes 32 and 44), no Bella Maria clones showed increased somy despite

similar levels of intra-chromosomal read-depth variation as clones from Cluster 2. Interestingly, most Bella Maria genomes showed severe reductions in sequencing coverage over chromosome 13. Such reductions did not occur in El Huayco or Ardanza (Fig. 6a). Somy plots for all initial samples are provided in Supplementary Fig. 14.

**Mysterious migrants imply further forms of genetic exchange.** Two samples in the dataset stand out as clear migrants with idiosyncratic genomic features that indicate the possibility of further genetic exchange events. TRT_3949 (sampled near El Huayco but associated to Cluster 1) and TCQ_3087 clones (sampled in Bella Maria but associated to Cluster 2) were the only samples for which geographic and nuclear phylogenetic neighbors did not match (Fig. 1, Supplementary Fig. 1,

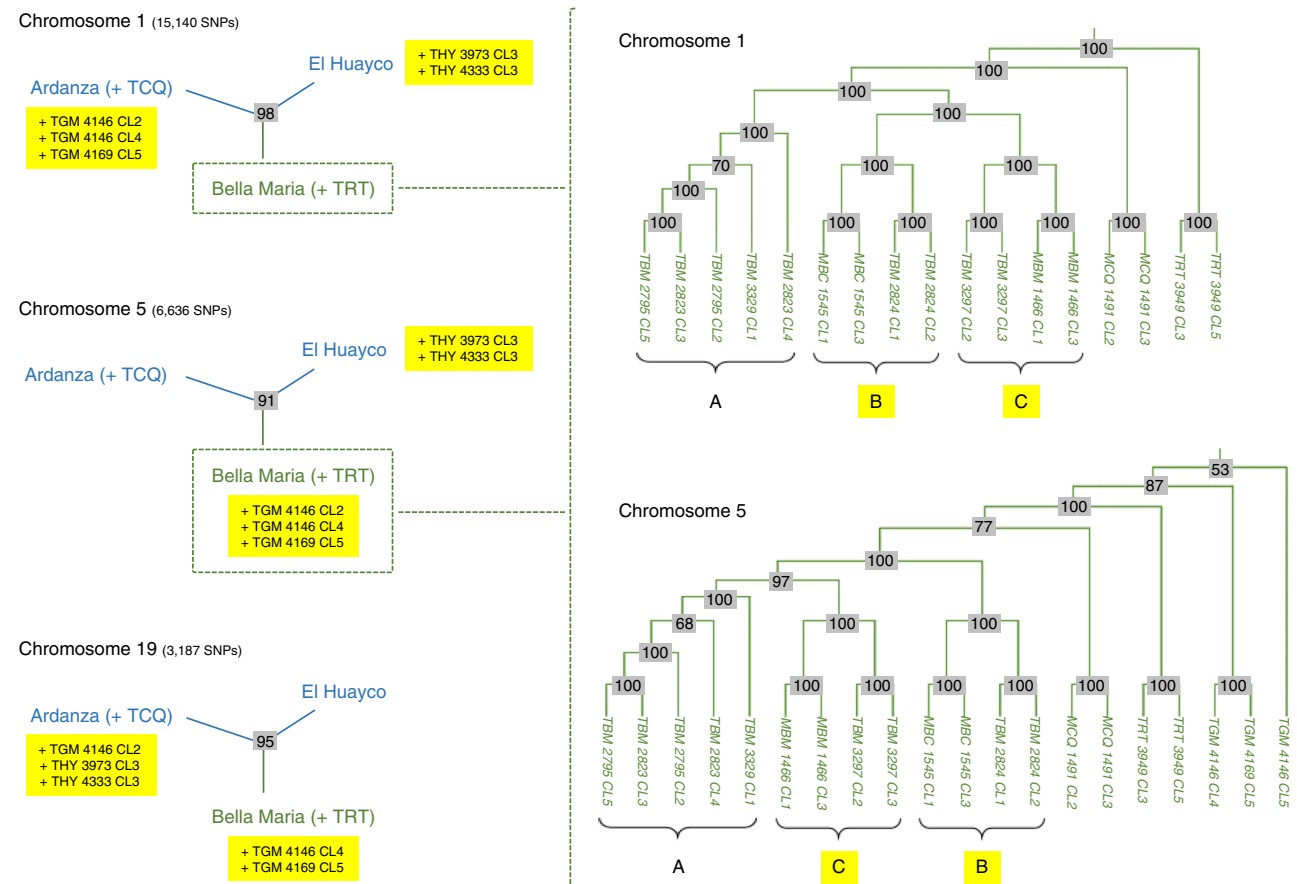

**Fig. 5** Incongruent trees exemplify independent chromosomal ancestries among *T. cruzi* I clones. Within individual sample genomes from Cluster 1 and Cluster 2, different chromosomes present different phylogenetic ancestries. For example, when neighbor-joining trees are constructed separately for chromosomes 1, 5, and 19, Gerinoma clones (prefix TGM) cluster with those from Ardanza on chromosome 1. On chromosomes 5 and 19, they cluster with clones from Bella Maria. El Huayco clones THY 3973 CL3 and THY 4333 CL3 also join the Ardanza clade on chromosome 19. Within Cluster 1 (right panel), chromosome 1 presents a monophyletic clade composed of MBC_1545 + TBM_2824 (labeled B) and MBM_1466 + TBM_3297 clones (C). TBM_2795 + TBM_2823 + TBM_3329 clones (A) form an outgroup. These clades rearrange on chromosome 5, where A changes places with C. The A + B monophylum occurs again on chromosome 19, while the B + C group makes appearances on chromosomes 9 and 16, etc. Discrepant phylogenies such as those highlighted here occur in various chromosomal comparisons throughout the genome. Nodes are labeled in gray with support values from 100 bootstrap replicates. Green denotes the Cluster 1 clade. Blue denotes Cluster 2. Yellow highlights unstable phylogenetic positions among different chromosomes. Branch lengths are not proportional to genetic distance

Supplementary Table 1). These clones also provided the dataset's only cases of discordant nuclear vs. mitochondrial phylogenies: TRT_3949 clones carried a maxicircle genotype otherwise found only in Cluster 2 and TCQ_3087 clones carried a maxicircle genotype highly divergent to any other observed in the study area (Supplementary Fig. 15a; see also *cytochrome b* alignment in Supplementary Fig. 15b). These apparent migrants were also exceptional in nuclear sequence alignment: within a single individual, some chromosome segments appeared to derive from Cluster 1, others from Cluster 2. For example, on chromosome 1, TCQ_3087 shared a heterozygous patch with Ardanza clones between ~785 and 920 kb. At ca. 1117 kb, sequences were similar to those of Gerinoma (TGM) clones and then, at ca. 1122 kb, similar to El Huayco clones. A long stretch of similarity to Cluster 1 ensued at ca. 1285 kb (Supplementary Fig. 9). TCQ_3087 and TRT_3949 clones were also the only samples for which homozygosity was widespread throughout the nuclear genome (Fig. 4a). Making up just 10% total polymorphic loci, bi-allelic SNPs were found restricted to scattered patches. High levels of overall homozygosity observed in these clones could not be attributed to certain chromosomes or to deviations in read-depth.

## Discussion

Our comparative genomic analysis of 45 biological clones from an area of endemic transmission supports the remarkable conclusion that (a) *T. cruzi* undergoes meiosis and (b) that grossly disparate reproductive strategies and rates of genetic exchange occur simultaneously at a single disease focus.

In a subsection of the region (Bella Maria), signs of regular meiotic sex are markedly clear. Genome-wide allele frequencies occur at Hardy–Weinberg equilibrium and ancestries among individuals fluctuate from chromosome to chromosome. Parasite genotypes on individual chromosomes appear equally mosaic: linkage between polymorphisms clearly correlates with map distance, disequilibrium plummeting within just a few hundreds of bp. We gauge that the meiosis driving these patterns of diversity occurs more than once every 1000 reproductive events in Bella Maria. Meanwhile, in nearby El Huayco, Ardanza and Gerinoma groups, meiosis appears essentially absent. Instead, these groups exhibit high levels of heterozygosity across the entire genome. We do detect discordant chromosomal phylogenies among these parasites, but recombination estimates within chromosomes match those for simulated, non-recombining controls and there are no signs of intra-chromosomal linkage decay. Alongside

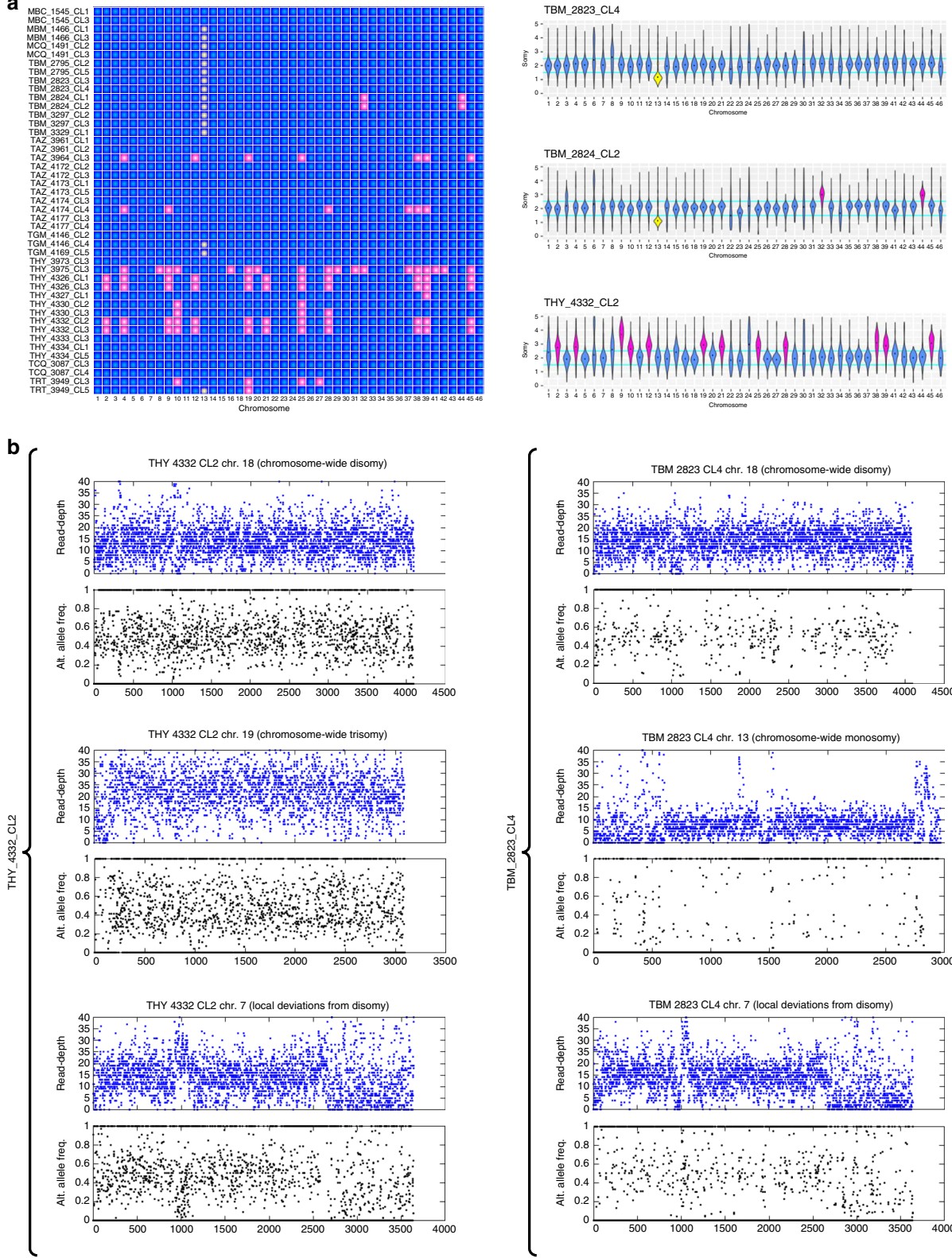

excess heterozygosity, several El Huayco and Ardanza clones also present extensive aneuploidy as well as long blocks of near-identical diplotypes.

The strong signatures of meiotic sex we report from Bella Maria redefine our understanding of *T. cruzi* biology and, alongside data from *T. b. brucei*[11] and *Leishmania*[17], indicate that this mode of genetic exchange is ancestral among medically important trypanosomatids[11]. We previously advised caution to those applying generalized theories of clonal evolution (e.g., PCE[26]) to parasitic protozoa[8]. Our revelations around *T. cruzi* population genomic structure in this study broadly support our case. Nonetheless, meiotic sex has never been observed in the

**Fig. 6** Group-level aneuploidy among *T. cruzi* I clones. **a** We distinguished chromosomal and intra-chromosomal copy number variation by evaluating sequence read-depth kernel density distributions. For example, these distributions suggest multiple cases of whole-chromosome somy elevation (highlighted in pink) for El Huayco clone THY_4332_CL2 (bottom right). Several clones from El Huayco and Ardanza present similar patterns (see Supplementary Fig. 14 for more density plots), as summarized in the heatmap at left. Read-depth densities suggest few cases of whole-chromosome somy elevation for clones from Bella Maria (e.g., TBM_2823_CL4 and TBM_2824_CL2 at right). However, mapping coverage drops dramatically (yellow) on chromosome 13 in most clones of this group. **b** Chromosome-wide shifts in sequence read-depth (blue) and alternate allele frequency (black) support whole-chromosome aneuploidies inferred from read-depth kernel density distributions above. In El Huayco clone THY_4332_CL2 (left column), for example, read-depth is elevated over the entirety of trisomic chromosome 19 (sequence positions are plotted on the x-axis). Alternate allele frequencies at heterozygous sites also distribute around values of 0.33 and 0.67 on this chromosome (as compared to frequencies around 0.50 on disomic chromosome 18). Cases of intra-chromosomal copy number variation for sample THY_4332_CL2 are marked by local shifts in read-depth and alternate allele frequency on chromosome 7. Comprehensive read-depth reduction on chromosome 13 is exemplified for Bella Maria clone TBM_2823_CL4 (right column). Alternate allele frequency values of 0 (indicative of the reference allele) predominate on this chromosome. Patterns on chromosomes 7 and 18 also point to intra-chromosomal copy number variation and stable disomy, respectively, for the TBM_2823_CL4 clone

laboratory and multiple aspects of meiosis in *T. cruzi* remain obscure[22]. The site of genetic exchange (vector or host) in *T. cruzi* is still not known, for example, nor is it understood from which parasite life cycle stage gametes might develop. In contrast, *T. b. brucei* gametes have been characterized in the salivary glands of tsetse flies and a mechanism for subsequent cytoplasmic fusion described[11]. Clearly much basic research remains to be done.

The distribution of genetic diversity we describe in Cluster 2 suggests that meiosis is largely absent among these strains. Patterns of heterozygosity recently observed in *T. b. gambiense* were attributed to the Meselson Effect[14,30], whereby mutations accumulate in the absence of recombination between homologous chromosomes during long-term clonality. The high levels of heterozygosity we observe in Cluster 2 differ in important ways from the *T. b. gambiense* dataset and from predictions of the Meselson Effect[14,30]. For example, discontinuities in genetic differentiation among individuals, instead of occurring as stretches of absolute homozygosity on disomic chromosomes as they did in *T. gambiense*[14], occur in our dataset as shared patches of heterozygosity among geographically distinct groups (e.g., El Huayco and Ardanza). Furthermore, we see no evidence of accumulation of private heterozygous sites within individuals as one might expect during long term asexual propagation – rather, over 50% of heterozygous sites are shared among samples in Cluster 2. If long-term asexuality is a poor explanation for heterozygosity in our dataset, an ancestral outcrossing event could perhaps have played a role.

In Cluster 2, we observed incongruent phylogenies between different chromosomes, but no evidence for linkage decay within individual chromosomes. In the only genetic exchange event observed experimentally in *T. cruzi* to date[22], parental genomes fused to tetraploid hybrids and then began erosion back toward the disomic state. This fusion-then-loss process resembles that in parasexual pathogenic fungi (*Candida* spp.) and allows for independent chromosomal ancestries without intra-chromosomal linkage decay[31]. Moreover, gene conversion in tetraploids can produce long tracts of increased identity on both homologs (i.e., the diplotype-sharing we refer to in our results) without loss of heterozygosity upon reduction to the disomic state[32]. This is especially true when genome erosion is biased against the retention of similar homologs[33], a condition that aligns with our results (e.g., we observed elevations to average homozygosity in just two chromosomes (Supplementary Fig. 16), not in fifteen (33%) as would be expected in the case of random chromosome loss). Aside from parasexual mating, however, polyploidization via failed meiotic division might also explain aneuploidy levels observed in El Huayco and Ardanza. Given that failed chromosome segregation typically involves failed crossover[34], this explanation also reconciles a lack of linkage decay in Cluster 2. A third possibility, high levels of aneuploidy via frequent

asymmetric chromosome allotment in mitotically dividing nuclei[35], also finds direct support in this dataset. Unlike occasional accounts of stable aneuploidy in the *Trypanosoma* genus[23,36,37], we detected short-term somy reductions in one of three re-sequenced aneuploid clones and also found evidence for sub-clonal ploidy variation, often termed mosaic aneuploidy in *Leishmania* research[35]. Congruent aneuploidies observed in closely-related Cluster 2 genotypes may thus reflect strain-specific or pre-adapted amplification programs as in *Leishmania* spp.[38,39].

The ecological and evolutionary drivers of distinct but sympatric reproductive modes in *T. cruzi* are not clear. While *T. cruzi* is able to infect a remarkable variety of insects and vertebrates, its stercorarian transmission route is highly inefficient. *T. cruzi*'s vectors and hosts vary immensely in transmission competence and availability and occupy an array of disparate niches (including the domestic-sylvatic interface)[40–42]. The parasite's life cycle thus likely represents a continuum of bottlenecks linked to frequent local extinction and recolonization events that increase levels of genetic drift and identity by descent (IBD). It may thus come to less surprise that observations of diffuse hybrid clonality around a restricted focus of sex in Bella Maria resemble spatio-temporal patterns of heterogony demonstrated in various other metapopulation systems[43–46]. Facultative sex often coincides with strong metapopulation structure, in which sexual variants are predicted to occupy core habitat (where population subdivision and inbreeding depression are minimized) while asexual variants disperse more freely without fitness costs from high IBD during frequent founding events[47]. Extensive asexual dispersal eventually brings divergent lineages into contact, creates potential to mate, form heterotic offspring, and reset clonal decay. Divergent homologs, however, may impair canonical sex when $F_1$ hybrids mate[48–50]. We noted mass elevation of Tajima's D in Cluster 2 of this study (Supplementary Fig. 17) and this offers further support for both hybridization and bottlenecked clonal propagation in generating an excess of intermediate-frequency variants over El Huayco and Ardanza[51,52]. Such excess, however, can also arise in simple (e.g., island model) demographic scenarios when mating becomes very scarce, whereupon the influence of demographic changes on the site-frequency spectrum becomes difficult to disentangle by current methods of inference[53]. Nevertheless, large patches of low differentiation observed in this study suggest a relatively recent contribution of hybridization on allelic divergence in El Huayco and Ardanza. Spatially correlated genetic substructures and low effective population sizes further substantiate the role of metapopulation dynamics in structuring genetic diversity in these groups.

Our genotype- and haplotype-based summaries of co-ancestry indicate that the meiotic parasite group in Bella Maria is genetically segregated from others with distinct reproductive histories in nearby El Huayco and Ardanza. Genetic discontinuity

occurs consistently for samples collected within a few kilometers distance and despite evidence for vector/host co-infection and migration between divergent groups. Putative migrants, possibly the progeny of these divergent groups, exhibit extensive (nuclear) homozygosity and, in the case of TCQ_3087 clones, extreme maxicircle divergence and very high maxicircle read-depth. Such observations are reminiscent of *L. major* crosses formed in non-native vectors[50] and of irregular, biparental mitochondrial inheritance in *T. b. brucei*[54].

Unexpected and poorly repeatable hybrid genomes have arisen on a number of occasions in experimental Tritryps research[17,22,55]. Sensitivity to cryptic biochemical cues is clearly high, but the molecular signals that incite recombination and control mating compatibilities within these species remain essentially unknown[56]. Our observations from the field do not identify such mechanisms but provide many relevant questions to explore. For instance, do ploidy barriers segregate transmission cycles in *T. cruzi*? Is certain monosomy (e.g., recall chromosome 13 in Bella Maria clones) associated with mating locus activation and sex? Is high homozygosity a direct result of improper mating or a subsequent effect (gene conversion, selfing, etc.)? What are the adaptive processes that underpin switching between different reproductive modes?

Our work presents hard evidence for meiotic sex in *T. cruzi*, as well as evidence for widespread clonal expansion, after episodic hybridization events. Recent evidence for sex obtained from Arequipa, Peru, in contrast, cannot be reliably distinguished from complex patterns of gene conversion in a fully clonal population[21]. Complex mating structures are of acute relevance to Chagas disease control. Recombination implies that important epidemiological traits are transferable, not locked into stable subdivisions in space and time (for case in point, consider, e.g., *SRA* gene transfer from *T. b. rhodesiense* to *T. b. brucei*[57]). Recombination has driven major changes in *T. cruzi* transmission in the past, including adaptation to the domestic niche[58,59]. Our data suggest that recombination may continue to transform contemporary disease cycles, as suggested for *Toxoplasma gondii*[60] and in *Leishmania* spp.[15,16,61]. The proven presence of a sexual cycle in *T. cruzi* should now reinvigorate the hunt for the site of genetic exchange within the host or vector, as well as its cytological mechanism. An in vitro model for meiotic genetic exchange in *T. cruzi* will dramatically improve our ability to distinguish the genetic bases of virulence, drug resistance and other epidemiologically relevant phenotypes. Determination of such traits may underpin future efforts to treat and control Chagas disease.

## Methods

**Parasite collection and cloning.** Trypanosomes were isolated from triatomines (*Rhodnius ecuadoriensis*, *Panstrongylus chinai*, *P. rufotuberculatus*, and *Triatoma carrioni*), rodents (*Rhipidomys leucodactylus*, *Simosciurus nebouxii*) and bats (*Artibeus fraterculus*) captured between 2011 and 2015 in eastern Loja Province, Ecuador. Capture coordinates, dates, and ecotypes (i.e., domestic, peri-domestic, or sylvatic) are provided in Supplementary Table 1 and associated protocols are detailed in previous studies led by the Center for Research on Health in Latin America (CISeAL)[62]. Individual parasite cells were cloned on solid medium to derive single-strain colonies following Yeo et al.[63]. Briefly, aliquots of $10^2$–$10^3$ epimastigote cells were mixed with 36 °C (molten) low melting point agarose and distributed over supplemented blood agar for stationary colony formation on petri dishes humidified with 5% $CO_2$ at 28 °C for ca. three months. Successful micro-colonies were then expanded in biphasic Novy-MacNeal-Nicolle (NNN) and Liver Infusion Tryptose (LIT) media. Complementary to 19 non-cloned primary cultures, this process yielded 64 axenic monocultures for subsequent DNA extraction and sequencing.

**DNA sequencing and variant discovery.** Genomic DNA was extracted from 83 *T. cruzi* cultures by isopropanol precipitation. DNA was sonicated and size-selected (median insert size = 198 nt; median absolute deviation = 69 nt) by covalent binding prior to paired-end sequencing on the Illumina HiSeq 2500 platform. To

guide variant discovery from resultant 125 nt sequence reads, we optimized reference-mapping and SNP-calling pipelines using paired-end Illumina reads (kindly provided by Carlos Talavera-López, SciLifeLab, Sweden) for *T. cruzi* TcI X10/1 (termed TcI-Sylvio elsewhere in the text) against the newly available PacBio sequence for the same reference strain[27]. Based on comparisons with TcI-Sylvio mapping results from various configurations in Smalt v.0.7.4 (we tested 12–14 kmer hash indexes and 2–8 base skip sizes), we chose to map samples using default settings (gap-open penalty = 6 and mismatch penalty = 4) in BWA-mem v0.7.3. We then sorted alignments with Samtools v0.1.18, marked PCR-duplicates with Picard v1.85 and identified single-nucleotide polymorphisms (SNPs) by local re-assembly with Genome Analysis Toolkit (GATK) v3.7.0[64] (also benchmarked for *L. donovani*[39]). Individual records produced by the HaplotypeCaller algorithm were subsequently merged for population-based genotype and likelihood assignment (GATK GenotypeGVCFs). Next, we calibrated variant filters by incrementally tightening thresholds for genotype quality (*Q*), read-depth (*D*) and local polymorphism density (*C*) until non-reference homozygous SNP-calls for TcI-Sylvio reached asymptotic decay. We then applied a virtual mappability (*V*) mask to exclude variant-calls in unreliable mapping areas of the reference genome. Specifically, we generated synthetic, non-overlapping 125 nt sequence reads from the PacBio assembly and mapped these back to itself with the Genomic Multi-tool software suite[65]. Only variants from areas with perfect, i.e., singleton (*V* = 1), synthetic mapping coverage were kept for analysis. Regions without singleton synthetic mapping coverage represented areas of low sequence complexity and/or redundancy and made up large fractions of all reference chromosomes (Supplementary Fig. 6a, b). With the above filters in place (*Q* > 1500; 10 > *D* < 100; *C* < 3 SNPs per 10 nt; *V* = 1), samples retained tens of thousands of homozygous variant loci, whereas TcI-Sylvio Illumina vs. TcI-Sylvio PacBio showed just 58. Nevertheless, the guide-sample presented ca. 20,000 small insertions and ca. 1,000 small deletions relative to the reference. We placed an additional mask ± 3 nt around these positions to avoid potential faults in the published genome. Final masking thus disqualified a total of 24 Mb (including all of chromosomes 17, 40 and 47) from polymorphism analysis. This highly conservative, diagnostic variant-screening approach also led us to exclude 24 low-depth samples for which genotypes could not be assigned at more than 40% variant sites. The final set of SNPs (in 59 samples) were annotated with snpEff v4.3t[66] using the TcI-Sylvio annotation file (http://tritrypdb.org/common/downloads/release-34/TcruziSylvioX10-1/gff/data).

**Computational phasing of heterozygous SNP sites.** Heterozygous SNP sites were phased over 30 iterations in BEAGLE v4.1[67]. The algorithm also imputes missing genotypes from identity-by-state segments found in the data. For haplotype co-ancestry and general comparative analysis, we restricted imputation to sites containing information for >60% samples. Later, in windowed phylogenetic comparison, however, we refrained from genotype imputation, i.e., used only sites with genotypes called in all individuals of the dataset.

**Detection of population genetic substructure.** We used the Neighbor-Net algorithm in SplitsTree v4[68] to visualize genome-wide phylogenetic relationships among samples in split network representation. Neighbor-Net extends Satou and Nei's neighbor-joining algorithm to accommodate evolutionary processes such as recombination and hybridization that lead to non-treelike patterns of inheritance. We also optimized a general time-reversible (GTR) substitution model with ascertainment bias correction (for accurate branch lengths in the absence of constant sites) to construct phylogenies from proportions of non-shared alleles, i.e., considering two haplotypes per variant site. Haplotype concatenations were also used to derive a minimum-spanning network, the set of edges that links nodes (individuals) by the shortest possible cumulative distance (i.e., maximum parsimony). We inferred genetic subdivisions in the sample-set by unsupervised k-means clustering and discriminant analysis of principle components (DAPC)[69]. These analyses applied genetic distances as the proportion of non-shared genotypes at all variant loci (i.e., considering variants at the genotypic level), as did Neighbor-Net and subsequent measurements of $F_{ST}$. After phasing heterozygous SNP sites (see above), we used fineSTRUCTURE v2.0.4[28] to recover traces of identity-by-descent in similar haplotypes. This program was recently used to expose hybridization events in congeneric *T. congolense*[12], as well as to disentangle reticulate ancestries in the closely-related *L. donovani* complex[39]. Its Chromopainter algorithm constructs a semi-parametric summarization of co-ancestry among all pairs of individuals based on variable rates of haplotype-sharing and linkage disequilibrium across sample genomes. We applied fineSTRUCTURE over a uniform recombination map, running $6 \times 10^5$ Markov chain Monte Carlo (MCMC) iterations ($1 \times 10^5$ iterations burn-in) and $4 \times 10^5$ maximization steps in the final tree-building step. Following indications of mosaic inheritance in these analyses, we assessed phylogenetic (dis)continuity by comparing genotype-trees built for individual chromosomes using neighbor-joining as implemented in ape v5.1[70] in R. We also built distance matrices based on haplotypes phased without imputation (see previous section) to quantify changes in genetic similarity between windows within chromosomes.

**Analyses of population genetic diversity and linkage.** To assess group-level genetic diversity, we calculated site-wise nucleotide diversity ($\pi$), Watterson's theta

($\theta$) and $F_{IS}$ using hierfstat v0.04-22[71] in R. $F_{IS}$ values rate heterozygosity observed within and between individuals, varying between -1 (all loci heterozygous for the same alleles) and 1 (all loci homozygous for different alleles). Values at 0 indicate Hardy–Weinberg equilibrium. We also measured rates of shared and private allele use (e.g., proportions of fixed heterozygous and singleton sites), assessed variant neutrality based on Tajima's $D$, quantified haplotype diversity by counting unique haplotypes per 10–100 kb, and scanned for long runs of homozygosity using VCFtools v0.1.13[72]. To determine linkage patterns within chromosomes 1, 5, 21, and 26 (the genome's best-mappable chromosomes – see Supplementary Fig. 6), we recoded sample genotypes with values of 0, 1, or 2 to represent the number of non-reference alleles at each variant site. After filtering out all SNP-pairs separated by masked sequence (in effect, confining analysis to sites separated by <100 kb), we measured linkage ($r^2$) as the correlation between genotypic allele counts and then binned $r^2$ into distance classes (from 0 to 100 kb in increments of 2 kb) to visualize relationships between map distance and linkage disequilibrium in R. These analyses were also run separately on core sequence areas, as defined by areas of synteny among TcI–Sylvio, *T. b. brucei* and *L. major* sequence assemblies annotated at http://tritrypdb.org. Intra-haplotypic recombination is unlikely to accompany meiotic crossover events in these areas of the genome[27]. Furthermore, we considered the extent to which our multiple-clone sampling strategy (chosen to avoid underrepresentation of SNP linkage (or diversity) within infections) might affect sample independence and variance-based statistical results. Linkage decay plots and other diversity metrics above were therefore also repeated using only one clone per infection source.

**Estimation of meiotic vs. mitotic division**. Following methods established to quantify complex microbial life cycles[73], we inferred the frequency of sex and clonality in *T. cruzi* isolates by comparing two different estimates of effective population size. The first estimate, $N_\rho$, is based on recombinational diversity observed in the sample. $N_\rho$ represents the number of cells derived from mating, i.e., the number of zygotes present in the population, and is calculated as $\rho/4r$ $(1 − F)$, where $\rho$ denotes nucleotide covariation between sites, $r$ denotes rate of recombination per bp per generation, and $F$ represents Wright's inbreeding coefficient. The second estimate, $N_\theta$, is based on mutational diversity observed in the sample. $N_\theta$ represents the total population size, i.e., the number of cells irrespective of sexual or mitotic origin, and is calculated as $\theta (1 + F)/4\mu$, where $\theta$ denotes nucleotide variation at single sites and $\mu$ denotes the rate of mutation per bp per generation. $N_\rho/N_\theta$ thus quantifies the frequency of meiotic reproduction in the population. To estimate this quotient from our sample, we derived $\theta$ from Watterson's estimator at non-coding sites and derived $\rho$ based on reversible-jump MCMC likelihood curves generated by the interval program in LDhat[74]. We used $1 \times 10^7$ MCMC iterations with 2000 updates between samples and block penalties set to five. We estimated $r$ from the equation $r = 0.043 \times S^{−1.310}$ and $\mu$ from the equation $\mu = 2.5866 \times 10^{10} \times S^{0.584}$. These regression models were developed in Rogers et al.[75] based on the observation that genome size ($S$) correlates strongly to rates of recombination and mutation in unicellular eukaryotes. We validated $\rho$ estimates by simulating input for LDhat in two ways. First, we created sequence alignment maps for ten non-recombinant individuals based on observed genotypes using BAMSurgeon v1.0.0[76]. Maps were set up for each individual by inserting fixed polymorphisms from the true sample set into TcI-Sylvio sequence reads, then spiking in random mutations at rates corresponding to the average number of pairwise differences in the observed data. Individual SNP records for the ten mutant alignment files were then compiled and merged in GATK as outlined above. In the second approach, we used fastsimcoal2 v5.2.8[77] to simulate ten non-recombinant and ten recombinant genotypes, applying $r$ and $\mu$ from above equations to an effective population of 100,000 diploid individuals under a finite-sites model of evolution for chromosome 1. We also visualized linkage patterns by measuring taxon topology weightings in windowed analysis. Taxon topology weightings provide a means to clarify phylogenetic structure by summarizing the extent to which tree topologies for a subset of samples contribute to the topology of the full tree[29]. We applied this concept to neighbor-joining trees constructed for overlapping 50 kb sequence windows in PhyML v3.1. Topology weightings were calculated and plotted across chromosomes with loess smoothing (span = 0.125) using scripts provided at GitHub repository https://github.com/simonhmartin/twisst. These analyses prompted further sequence visualizations with Artemis v.16.0.0[78] genome browser tool.

**Chromosomal somy analysis**. To estimate somy levels for each sample, we first measured mean read-depth for successive 1 kb windows spanning each chromosome. We then calculated the median of these windowed depth-means ($m$), i.e., a median-of-means ($M_m$), for each chromosome. After testing at various distribution points, we let the 30th percentile (p30) of (skewed) $M_m$ values represent expectations for the disomic state, estimating copy number for each chromosome by dividing its $M_m$ by the sample's p30 value and multiplying by two. This procedure produced estimates of disomy for all chromosomes of the TcI–Sylvio guide-sample and outperformed techniques based on different window-sizes as well as those refined according to sequence annotation (e.g., only single-copy genes) or mapping quality (data not shown). We validated cases of chromosomal copy number variation by plotting kernel densities for $m$, as well as by assessing raw depth and alternate allele frequencies across variant sites. True,

whole-chromosomal trisomy, for example, should translate to chromosome-wide elevations in read-depth and reductions in minor allele contributions to ca. 33% (i.e., one B and two A alleles – and, in cases of tri-allelism, one of each A, B, and C alleles) at all heterozygous (i.e., A/B/B or A/B/C) sites. Intra-chromosomal amplification, in contrast, should create local shifts in read-depth and allelic composition within chromosomes. In follow-up assessment of temporal and sub-clonal ploidy variation, we re-sequenced three clones and derivative subclones on the Illumina NextSeq 500 platform (Supplementary Table 4). Subclones were obtained using the limiting dilution method as described in Messenger et al. (section 3.2.3)[79]. Briefly, logarithmic phase cell cultures were diluted to 50 parasites/ml in Roswell Park Memorial Institute (RPMI) 1640 medium, then divided into 200 µl aliquots across multiple 96-microwell plates. Wells presenting individual cells were incubated at 28 °C for ca. 6 weeks and further expanded in LIT.

**Reporting Summary**. Further information on research design is available in the Nature Research Reporting Summary linked to this article.

## Data availability

Sequence data that support the findings of this study are available at Sequence Read Archive (SRA) BioProject PRJNA552612. All other relevant data are available from the corresponding author on reasonable request.

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

## Acknowledgements

Special thanks to A. Vela, C. Cilveti, C. García, H. Darban, J. Galbraith and S. Ocaña-Mayorga for help with laboratory procedures. Thanks also to A. MacLeod, D. Streicker, J.-C. Dujardin, R. Biek and W. Weir for advice in data analysis and report. This study was funded by the Division of Microbiology and Infectious Diseases, the National Institute of Allergy and Infectious Diseases and the National Institutes of Health (DMID/NIAID/NIH grants AI077896-01 and AI105749-01A1); the NIH-Fogarty Global Infectious Disease Training Program (grant TW008261); the Pontifical Catholic University of Ecuador (I13048, J13033, K13063 and L13225); a Wellcome Trust Institutional Strategic Support Fund secondment award 204820/Z/16/Z, a Medical Research council award MR/M026353/1, and the Scottish Universities Life Sciences Alliance. Work with *T. cruzi* was conducted under framework agreement No. MAE-DNB-CM-2015-0030 from the Ecuadorian Ministry of Environment. The funders had no role in study design, data collection and analysis, decision to publish, or preparation of the manuscript.

## Author contributions

P.S., M.S.L., M.A.M. and M.J.G. conceived the study. P.S., J.A.C., J.M.-S. and B.A. conducted laboratory procedures. P.S., H.I., F.V.D.B. and M.S.L. conducted data analyses. P.S. and M.S.L. wrote the manuscript. All authors critically reviewed and approved the manuscript.

## Additional information

**Competing interests:** The authors declare no competing interests.

