## [Peer Review File · Nature Communications]

Reviewers' comments:

Reviewer #1 (Remarks to the Author):

This article concerns the population structure of *Trypanosoma cruzi* and addresses the controversial questions whether it has sexual reproduction, and if so, how frequently. *T. cruzi* is an important human parasite and Chagas disease is a neglected tropical disease, both factors that potentially make this a significant study. Moreover, hybrid strains of *T. cruzi* from human patients have been described, showing the importance of genetic exchange even if a rare event. The authors have genome sequenced *T. cruzi* collected from triatomine bugs (mainly) or mammalian hosts (3 isolates from rodents and bat) in a focus of Chagas disease in Ecuador. Some isolates were cloned, some not and appear to be mixed infections. SNP analysis against reference *T. cruzi* genome divides the isolates into 2 genetically distinct groups, though close in space and time, even to the extent of being isolated from the same host on occasion. Further investigation of the 2 groups showed that the population structure was different – one group appeared to be random mating (no deviation from H-W equilibrium), while the other appeared clonal with fixed heterozygosity. The main conclusion the authors draw is that *T. cruzi* has meiotic sex under certain circumstances, contrary to current views that *T. cruzi* is clonal or has some form of genetic fusion that creates hybrids. Details of sexual reproduction in *T. cruzi* remain to be worked out, but the paper at least provides evidence that the process takes place.

I might have found the paper more convincing if the data were explained more clearly and have some queries on the numbers of isolates/clones analysed, the accuracy of terminology and interpretation as detailed below.

Line 48: is it true that frequent meiotic recombination been established in *T. b. brucei* and if so, how frequent is it?

Line 71: "high frequency via a mechanism consistent with classic meiosis" - how frequent is high frequency? what aspects of meiosis – crossing over and recombination between homologues? Reassortment and segregation?

Line 72: what definition of a population is being used here? There are 2 genetically distinct clusters, but this is within the population of trypanosomes from this focus. Same line 100-101 – if parasites co-occur in the same vectors and hosts, this is one population.

Line 74/75: "significant contribution towards the consolidation of current theories .." perhaps the authors could expand on what they mean here.

Line 85: what about the other 2 clones? Numbers don't seem to add up – 45 clones are mentioned at start of this section, but $17+11+12+3 = 43$. Terminology confused – both clones and isolates. Overall, the number of individual hosts sampled from each area is quite small – because a number of clones were examined from each host, this has increased the sample size. However, clones from the same host are not independent samples – indeed they often seem to be genetically related – so this is not a random sample from the *T. cruzi* population in this area. How does this factor into the population genetics stats?

Line 97-99: useful to mention the strain names here – the table listing the isolates/clones is in Supp. Table 1.

line 99-100: not sure what is meant here – were the 4 non-cloned isolates composed of different genotypes, revealed after cloning? These clones are not shown in Fig 2, so there is no evidence for this statement.

Line 104: 15 BM isolates – should this be clones?

Line 123: "short blocks of identity consistent with meiotic re-assortment" - not recombination?

Line 127: crossing over occurs between homologous chromosomes during meiosis.

Line 148: "sexual mating" = sexual reproduction?

Line 156: "some degree of genetic exchange" – meaning?

Line 209: might want to rephrase this in case it's taken out of context!

Line 224: PCE model doesn't rule out sexual reproduction.

Line 226: "meiotic exchange" = sexual reproduction?

Line 238: "homozygosis" – homozygosity?

Line 271: "eclectic but stercorearian" – explain what is meant.

Fig 2 legend – horizontal banding in 3 or 4 isolates? Legend contradicts what is shown in fig2 and stated in text. What is the key to colour in Fig 2 heatmap and why are some squares black or white?

Reviewer #2 (Remarks to the Author):

The paper “Meiotic and non-Mendelian genetic exchange in *Trypanosoma cruzi*, agent of Chagas disease” is certainly provocative, and is the first study to perform a comparative phylogenomic analysis on the “core” genome using field isolates of the TcI subtype of *T. cruzi* collected from an endemic focus of infection in Ecuador. Until recently, the *T. cruzi* genome assembly was highly fragmented, making it impossible to perform genome-wide comparative phylogenomic analyses with any confidence. Taking advantage of the recently assembled genome of the TcI Sylvio strain that is resolved to 47 scaffolds, Schwabl et al. assessed whether *T. cruzi* clones derived from a collection of 26 strains isolated predominantly from insects, bats and rodents are expanding clonally, or if they had marks of genetic hybridization within their “core” genome assembly, which is comprised of ~17Mb of sequence. They concluded that two populations of *T. cruzi* exist, one that is in Hardy-Weinberg equilibrium and is undergoing genetic exchange consistent with a meiotic process, whereas the other is comprised of admixed lines that appear to be genomic mosaics. They propose that nuclear fusion in a manner consistent with a parasexual process is the most likely explanation for the origin of these latter strains. While this is a compelling idea, other mechanisms that could equally explain the origin of these hybrid strains are either discredited, or not appropriately examined, and key references that advance karyotype instability and haplotype selection as a mechanism to explain genome evolution within kinetoplastids are not included. While the authors present an impressive dataset of sequences analyzed in a number of meaningful ways, they advance only one process to explain the non-Mendelian genetic mosaicism observed, at the expense of other possible mechanisms, that could be tested to give more confidence in their interpretation. Other assumptions made and several aspects of the data analyses performed need to be better developed before they advance the conclusions they draw, and these points are raised in the specific comments section.

The manuscript is impressively well written. As stated above, the authors advance an interpretation that may not be consistent with the data, so several additional analyses should be performed in order to substantiate the mechanistic statements advanced in the manuscript. Specifically, the claim that their analyses identified “widespread clonal expansion from non-Mendelian mating events” [see lines 313-314]. Which they go on to argue is consistent with a “fusion-then-loss process resembles that in parasexual pathogenic fungi (*Candida* spp.) and allows for independent chromosomal ancestries without intra-chromosomal linkage decay” [see line 248-250]. While this is a compelling idea, the dataset is static, and exists as a snapshot in time. The origin of the genomic mosaicism as parasexual is only inferred from their comparative phylogenomic analyses, but equally could arise by haplotype selection and mosaic aneuploidy, the result of karyotype instability and genome plasticity that has been well-described in the related kinetoplastid *Leishmania*. As this exists as a plausible alternative explanation to their findings, especially since the absolute number of SNPs resolved is actually quite limited (often less than 0.1%). Mosaic aneuploidy is a common occurrence during in vitro cloning of *Leishmania* and would generate seemingly independent chromosomal ancestries without intrachromosomal linkage decay. And relatively straightforward experiments could be performed to test for differences in somy, a signature of a parasexual process, that rather exist as the result of cloning during in vitro culture. While it is not readily possible to test for a parasexual loss of chromosomal somy (and no tetraploid intermediates have been identified in their dataset), it is possible to take a parent isolate, clone it, propagate it, then sub-clone it to test for differences in somy, which may very well explain the current data shown. In fact, the 3 parents [THY_4327, MCQ_1491, TAZ_4177] that did have aneuploid clones associated with them were generally disomic in the Supplemental Dataset and may support haplotype selection to explain their origin [see Barja et al, 2017. *Nature Ecol Evol*].

Furthermore, 7 other parent populations were presented in Figure 2, but did not have clones associated with them {THY_4328, THY_3978, THY_4331, TBM_2798, TGM_4157, TAZ_4179, TAZ_3968}. What was the rationale for excluding parent populations from the population genomic analyses when the fineSTRUCTURE patterns did not suggest they were mixed infections. Interestingly, the two clones from MCQ_1491 were in fact genetic recombinants (see Fig 1a) and chromosome painting at core genome resolution of the parent against these two clones would likely provide important insight into the genetic mechanisms at play. For example, do blocks of heterozygosity exist in the parent, that are phased into discrete haplotypes in the "recombinant" clones?

Specific comments:

1 Both the Neighbour-Net and Fine Structure plots show evidence of genetic recombination and shared ancestry, but not their position or coverage within the core genome, to support the finding that these parasites undergo classical meiotic reproduction. Fine Structure suggests ~10 distinct ancestries across the two populations, how have these distinct ancestries been inherited genome-wide? Visualizing the extent of this, across the different strains examined would be a powerful figure to show the mosaic nature and the frequency of identifying different admixtures within the population genetics of this parasite. It is not currently clear whether admixture has occurred in the majority of strains - or just relatively few. Failure to show positionally across each genome the number of ancestries that exist, and whether/how they have recombined across the phased genomes is a deficit in the manuscript. Indeed, visualization tools exist that have been employed in a number of papers to facilitate depicting the types of relationships that are being advocated herein. See for example Rogers et al, PLoS Genetics, 2014 Fig 5, or Bajra et al, Nature Ecol Evol, 2017 Fig 4. Likewise, inclusion of a bottlebrush plot, if number of ancestries is limited, would be an informative pair-wise comparison that could be generated genome-wide, to identify the extent to which blocks of heterozygosity are consistent with a meiotic/genome-fusion process.

2. In Figures 3c and 3d, the pairwise SNP association plots establish that the populations in EH and AR are not in linkage disequilibrium along chromosome 1, which is supported by the reticulate patterns identified in the Neighbour-Net plot in Fig 1a. The structuring in the pattern of SNP variant-associations is consistent with recombination of a limited number of alleles across the strains examined, underpinning why the chromosome painting analysis above should be performed. What is clear is that the SNP density is relatively low (see Figure 5b), and the patchy heterozygosity observed in the populations outside BM could be consistent with an asexual expansion model, so additional analyses proposed should be performed to better inform the exact phylogenomic relationship that exists between strains in this region.

3. Figure 6 is quite difficult to interpret. It would appear that MBM_1466 shares regions of the same haplotype as TRT_3949, but differs in heterozygosity in small blocks on chromosome 5. Could this be the result of random mutation consistent with genetic drift? And that TAZ_4173 and THY_4334 likewise share similar haplotypes, but differ in the extent of their heterozygosity, but are conserved relative to others shown. It is therefore important to see this analysis done genome-wide. Is it possible to perform this using concentric tracks, depicting heterozygosity in one track, and chromosome painting on the other?

4. This entire analysis would benefit from showing a genome-wide SNP density plot, to formally show that the SNP density is low, and to more readily highlight regions of increased SNP density that likely represent introgression from distinct genetic ancestries that have undergone an independent evolution. This is particularly relevant for the strains outside of BM, as this analysis may help to shed light on whether the relationship among strains is one of mutational drift by asexual expansion, or introgression of a limited number of haplotypes followed by haplotype selection/gene conversion to a homozygous state.

5. Is it known whether *T. cruzi* is prone to mitotic recombination, in similar to what is seen in the

var genes for *P. falciparum*? The pair-wise SNP density plots, overlaid with gene model information may be informative to determine if the biallelic hot-spots may represent gene family expansions, which could produce regions of patchy heterozygosity that appear less-random.

6. Fig S11, No outgroup for A is depicted – the figure would be more meaningful if additional reference strains were added, including TcI Sylvio, TcII Esmeraldo, TcVI Brener? In Fig S11b, CytC – the analysis is incomplete. Where do the other strains analysed from Ecuador cluster – with Tc1, to distinguish just how different 3087 is?

Response letter to Reviewers 1 and 2

We thank both reviewers very much for their careful examination of our work. The comments and queries have been immensely helpful in improving the clarity of our manuscript. **Crucially, the reviewers have helped us focus the manuscript on its primary, and highly consequential message – our discovery of panmictic population genetic structure consistent with frequent meiotic genetic exchange in natural *T. cruzi* populations.** As detailed in individual responses to Reviewer 1 and 2, we now thoroughly describe the frequency and distribution of recombination in our dataset both within individual genomes and across multiple clones, using illustrative SNP polymorphism ‘painting’ methods as well as quantitative phylogenetic approaches. As in the original manuscript, we also commit attention to a secondary, yet likewise very epidemiologically relevant finding of the study – the observation that *T. cruzi* groups with divergent reproductive modes coexist at the same transmission focus. Specifically, we find ‘Cluster 2’ genomes with pervasive, yet predominantly non-private genome-wide heterozygosity more consistent with ancestral hybridization than independent, Meselson-like accumulation of SNPs. In the first submission, however, we added speculation that this hybridization could have constituted a non-meiotic polyploid hybridization event – consistent with that observed from experimental crosses in the laboratory (Gaunt et al 2003, Nature). This additional point ballooned into an unnecessary distraction from the main message of the paper. Our arguments for polyploidization were based heavily on the observation of congruent trisomies in multiple samples (i.e., similar sets of chromosomes amplified in different clones). We have now repeated some analyses on three re-sequenced (130M Illumina NextSeq) aneuploid clones, also their subclones, to address the possibility of short-term and/or sub-clonal karyotypic instability in the sample set. We found evidence for both. These indications of frequent mitosis-driven karyotype variation do not exclude the possibility of past polyploidization, but have led us to now cut this speculation almost entirely from the manuscript. The hypothesis that closely-related Cluster 2 genomes exhibit a similar karyotypic plasticity over generations is more parsimonious, especially given various precedent from *Leishmania* research. Key references are provided in the text.

More detailed responses follow to individual reviewer comments (grey shading) below.

Reviewer #1 (Remarks to the Author):

This article concerns the population structure of *Trypanosoma cruzi* and addresses the controversial questions whether it has sexual reproduction, and if so, how frequently. *T. cruzi* is an important human parasite and Chagas disease is a neglected tropical disease, both factors that potentially make this a significant study. Moreover, hybrid strains of *T. cruzi* from human patients have been described, showing the importance of genetic exchange even if a rare event. The authors have genome sequenced *T. cruzi* collected from triatomine bugs (mainly) or mammalian hosts (3 isolates from rodents and bat) in a focus of Chagas disease in Ecuador. Some isolates were cloned, some not and appear to be mixed infections. SNP analysis against reference *T. cruzi* genome divides the isolates into 2 genetically distinct groups, though close in space and time, even to the extent of being isolated from the same host on occasion. Further investigation of the 2 groups showed that the population structure was different – one group appeared to be random mating (no deviation from H-W equilibrium), while the other appeared clonal with fixed heterozygosity. The main conclusion the authors draw is that *T. cruzi* has meiotic sex under certain circumstances, contrary to current views that *T. cruzi* is clonal or has some form of genetic fusion that creates hybrids. Details of sexual reproduction in *T. cruzi* remain to be worked out, but the paper at least provides evidence that the process takes place. I might have found the paper more convincing if the data were explained more clearly and have some queries on the numbers of isolates/clones analysed, the accuracy of terminology and interpretation as detailed below.

Many thanks to Reviewer 1 for the careful examination of our work. Our responses follow below. We have made extensive revisions to solve all technical and terminological ambiguities the reviewer identified in the manuscript.

*Line 48: is it true that frequent meiotic recombination been established in *T. b. brucei* and if so, how frequent is it?*

Meiosis is recognized as a normal part of the developmental cycle of *T. brucei* ssp. *brucei* in the tsetse fly. Peacock et al. (Curr. Biol. 2014) have repeatedly identified haploid cells that display the behavior expected of a gamete. When these cells are mixed, they readily form pairs or clusters and undergo cytoplasmic fusion. In a follow-up study to examine the frequency of hybrid trypanosome production, yellow-fluorescing hybrid trypanosomes occurred in 75% of doubly-infected tsetse salivary glands – not only did mating occur, it occurred at ‘high frequency’ (Peacock et al. 2016, Parasites and Vectors). We have now included this extra detail in lines 50-52, and have changed “frequent” to “regular” in order to avoid any possible misleading in line 49.

Line 71: “high frequency via a mechanism consistent with classic meiosis” - how frequent is high frequency? what aspects of meiosis – crossing over and recombination between homologues? Reassortment and segregation?

We first estimated crossover frequency (Morgans/kb) by coalescent theory in LDhat, then used these values to estimate effective population size (N_p). By comparing N_p to a second estimate of population size based on mutational diversity, we obtained a quotient for the number of cells derived from mating relative to the total number of cells, irrespective of sexual or mitotic origin. Crossover frequency values (ρ) for Bella Maria suggested ca. 3 meioses per 1,000 mitotic events in this group (lines 144-151).

Apart from this aspect of meiosis – genetic recombination between homologous chromosomes, consequences of which are also illustrated plots of linkage decay (Fig. 3) and intra-chromosomal phylogenetic instability (Twisst results, Fig. 4) – we highlight several further aspects/symptoms of meiosis in this manuscript, including chromosomal reassortment, Mendelian allelic distributions, disomic karyotypes. Please see also new genome-wide SNP-painting in Fig. S9. We emphasize that these are the specific aspects that lead to the conclusion of sexual population structure in our study in the summary paragraph at the beginning of the discussion section – lines 246-257.

Line 72: what definition of a population is being used here? There are 2 genetically distinct clusters, but this is within the population of trypanosomes from this focus. Same line 100-101 – if parasites co-occur in the same vectors and hosts, this is one population.

Indeed, we had also debated about this point. The population term is traditionally used in the context of canonical sexual organisms inhabiting a particular geographic area. Individuals of a certain ‘population’ are considered more likely to breed with each other than with individuals of another ‘population’. In the context of facultative sexual/clonal reproduction and meta-population structure, this concept becomes more difficult to apply. Therefore, we have now opted for a more neutral term, ‘group’ throughout the manuscript.

Line 74/75: “significant contribution towards the consolidation of current theories ..” perhaps the authors could expand on what they mean here.

We have edited for clarity on lines 77-80.

Line 85: what about the other 2 clones? Numbers don’t seem to add up – 45 clones are mentioned at start of this section, but $17+11+12+3 = 43$. Terminology confused – both clones and isolates.

Indeed, we used these terms as synonyms, alluding to the fact that ‘plate-cloning’ is essentially ‘isolating’ single-strain cells from a potentially mixed-strain primary infection. We have amended this, now using only the term ‘clone’ for plate-clones. With this clarification, we hope the sum of 45 clones is now evident:

- 15 parasite clones were isolated from animals captured in Bella Maria and also clustered together phylogenetically based on this common origin
- 11 parasite clones were isolated from animals captured in Ardanza and also clustered together phylogenetically based on this common origin
- 12 parasite clones were isolated from animals captured in El Huayco and also clustered together phylogenetically based on this common origin
- 3 parasite clones were isolated from animals captured in Gerinoma and also clustered together phylogenetically based on this common origin
- 2 parasite clones were isolated from animals captured in Bella Maria but showed aberrant phylogenetic positions (etc.)
- 2 parasite clones isolated from animals captured near El Huayco but showed aberrant phylogenetic positions (etc.)

*Overall, the number of individual hosts sampled from each area is quite small – because a number of clones were examined from each host, this has increased the sample size. However, clones from the same host are not independent samples – indeed they often seem to be genetically related – so this is not a random sample from the *T. cruzi* population in this area. How does this factor into the population genetics stats?*

We also find this a very interesting matter. What constitutes a non-pseudo-replicative sample that meaningfully describes facultative clonal/sexual behavior in an organism that occurs both in homogeneous and multiclonal infections distributed across different yet interacting vectors and hosts? Would exceedingly independent sample-taking, for example only once per infection, possibly also only from seldom interacting vectors/hosts (e.g., not from

the same house, tree etc.) and from a bottleneck-free population, best represent the way parasite diversity distributes in the landscape, and best serve to gauge genetic connectivity vs. segregation, etc? Or rather, might such data collection inflate linkage equilibrium and nucleotide diversity (etc.), dampen traces of genetic exchange and miss interesting instances of divergent ancestries present within single infections? We believe there is no single best model data set / sampling configuration for population genetic inference; that is, there is no single calibration point against which to normalize our results. In our setting, we find our multiple within-host sampling important because, apart from acknowledging replicate/near-replicate observations as a proper biological representation of the study organism, it also increases the chance to apprehend direct recombinant relatives in the data, and it enhances observation of genetic diversity at the micro-geographic scale, for example, demonstrating diversity within hosts/vectors in relation to diversity among hosts/vectors across the landscape. In this way, and against expectation perhaps, we found that pairwise similarity does not always reflect common host/vector or geographic source. On a number of occasions, a parasite clone appears more similar to clones from another infection than to those within its own vector/host (e.g., TBM_2823_CL3 or TAZ_4172_CL2 or TAZ_4173_CL5). Samples taken from the same infection are indeed in most cases highly similar, but this is a true representation of diversity at the landscape scale. We do not consider it pseudo-replication *per se*, but yes of course, this sampling design will affect population genetic results to a certain extent.

We evaluated the extent of this effect by recalculating all population genetic metrics in Tbl. 1 and repeating linkage decay analysis with only one representative clone per infection. Relative differences and overall inference did not change. Please see Appendix of this letter. Relative and absolute values are very similar, except for nucleotide diversity (π) which, unsurprisingly, becomes elevated upon removal of all more closely related individuals (App. Tbl. R1). The omission also leads to a misleading statistical artefact for the Ardanza group regarding percent SNPs in Hardy-Weinberg Equilibrium (HWE). When statistically testing for deviations from HWE, small sample sizes begin to prohibit correct rejection of the null-hypothesis (HWE). We demonstrated this by creating artificial datasets of purely clonal individuals with N=2, N=3, N=4, N=5, N=6, N=7, N=8, N=9 and N=10, for which null-hypothesis rejection fails at N \leq 6 (App. Fig. R1). Please see also linkage decay re-plotted with only one clone per infection in App. Fig. R2. The curves completely mirror initial results (Fig. 3). Given these congruences, we do not think it is necessary to provide above validations in the new manuscript text itself.

Line 97-99: useful to mention the strain names here – the table listing the isolates/clones is in Supp. Table 1.

Thank you, yes, done.

line 99-100: not sure what is meant here – were the 4 non-cloned isolates composed of different genotypes, revealed after cloning? These clones are not shown in Fig 2, so there is no evidence for this statement.

It is inferred. We now use “may” and “possible” to indicate speculation (lines 26 and 106).

Line 104: 15 BM isolates – should this be clones?

Yes, we used terms ‘isolates’ and ‘clones’ synonymously in reference to plate-clones, never using these terms for non-cloned samples. But this was obscure, and we now use only the term ‘clone’ for clones.

Line 123: “short blocks of identity consistent with meiotic re-assortment” - not recombination?

Great catch, thank you, we have amended to “Short blocks of shared identity among samples could be consistent with meiotic recombination” (lines 127-128).

Line 127: crossing over occurs between homologous chromosomes during meiosis.

Yes, we should have specified ‘homologous’, as specified now (line 132).

Line 148: “sexual mating” = sexual reproduction?

Yes, we consider sexual reproduction as successful sexual mating. However, mating is more often used at the macro-ecological level, especially to describe behaviour, such that in the molecular context here ‘reproduction’ is the better term. We have changed this now (line 167).

Line 156: “some degree of genetic exchange” – meaning?

Genetic exchange has occurred, even if long ago (for example common hybridization event discussed further on) –

now modified in lines 177-179.

Line 209: might want to rephrase this in case it's taken out of context!

Alright, we have modified to “signs of regular meiotic” (line 246).

Line 224: PCE model doesn't rule out sexual reproduction.

Yes, but we believe that the intricate and distinctive biologies coming to light for different trypanosomatids may be better explained without a necessary overarching dogma set up with limited information decades ago.

Line 226: “meiotic exchange” = sexual reproduction?

Yes. We have changed the term to “meiotic sex” to improve clarity (line 258).

Line 238: “homozygosis” – homozygosity?

Yes, we used the suffix -osis because it implies the *process* of creating homozygous segments, i.e., gene conversion, but it sounds weird indeed. Thanks, we have changed it (line 275).

Line 271: “eclectic but stercorarian” – explain what is meant.

We mean wide host/vector and habitat range, but ineffective (bottlenecking) transmission. Now clarified in lines 304 - 305.

Fig 2 legend – horizontal banding in 3 or 4 isolates?

Typo! Thank you, sorry this should say four original (non-cloned) infections. Corrected now.

Legend contradicts what is shown in fig2 and stated in text. What is the key to colour in Fig 2 heatmap and why are some squares black or white?

Yes, we should have included the color scale. It is included now. Black is the maximum color intensity and indicates sample pairs with total haplo-segment similarity (number of times each other's segments rank as ‘nearest neighbors’ in FineStructure analysis) much closer than to any other sample.

Reviewer #2 (Remarks to the Author):

The paper “Meiotic and non-Mendelian genetic exchange in *Trypanosoma cruzi*, agent of Chagas disease” is certainly provocative, and is the first study to perform a comparative phylogenomic analysis on the “core” genome using field isolates of the TcI subtype of *T. cruzi* collected from an endemic focus of infection in Ecuador. Until recently, the *T. cruzi* genome assembly was highly fragmented, making it impossible to perform genome-wide comparative phylogenomic analyses with any confidence. Taking advantage of the recently assembled genome of the TcI Sylvio strain that is resolved to 47 scaffolds, Schwabl et al. assessed whether *T. cruzi* clones derived from a collection of 26 strains isolated predominantly from insects, bats and rodents are expanding clonally, or if they had marks of genetic hybridization within their “core” genome assembly, which is comprised of ~17Mb of sequence. They concluded that two populations of *T. cruzi* exist, one that is in Hardy-Weinberg equilibrium and is undergoing genetic exchange consistent with a meiotic process, whereas the other is comprised of admixed lines that appear to be genomic mosaics. They propose that nuclear fusion in a manner consistent with a parasexual process is the most likely explanation for the origin of these latter strains. While this is a compelling idea, other mechanisms that could equally explain the origin of these hybrid strains are either discredited, or not appropriately examined, and key references that advance karyotype instability and haplotype selection as a mechanism to explain genome evolution within kinetoplastids are not included. While the authors present an impressive dataset of sequences analyzed in a number of meaningful ways, they advance only one process to explain the non-Mendelian genetic mosaicism observed, at the expense of other possible mechanisms, that could be tested to give more confidence in their interpretation. Other assumptions made and several aspects of the data analyses performed need to be better developed before they advance the conclusions they draw, and these points are raised in the specific comments section.

The manuscript is impressively well written. As stated above, the authors advance an interpretation that may not be consistent with the data, so several additional analyses should be performed in order to substantiate the mechanistic statements advanced in the manuscript. Specifically, the claim that their analyses identified “widespread clonal expansion from non-Mendelian mating events” [see lines 313-314]. Which they go on to argue is consistent with a “fusion-then-loss process resembles that in parasexual pathogenic fungi (*Candida* spp.) and allows for independent chromosomal ancestries without intra-chromosomal linkage decay” [see line 248-250]. While this is a compelling idea, the dataset is static, and exists as a snapshot in time. The origin of the genomic mosaicism as parasexual is only inferred from their comparative phylogenomic analyses, but equally could arise by haplotype selection and mosaic aneuploidy, the result of karyotype instability and genome plasticity that has been well-described in the related kinetoplastid *Leishmania*. As this exists as a plausible alternative explanation to their findings, especially since the absolute number of SNPs resolved is actually quite limited (often less than 0.1%). Mosaic aneuploidy is a common occurrence during *in vitro* cloning of *Leishmania* and would generate seemingly independent chromosomal ancestries without intrachromosomal linkage decay. And relatively straightforward experiments could be performed to test for differences in somy, a signature of a parasexual process, that rather exist as the result of cloning during *in vitro* culture. While it is not readily possible to test for a parasexual loss of chromosomal somy (and no tetraploid intermediates have been identified in their dataset), it is possible to take a parent isolate, clone it, propagate it, then sub-clone it to test for differences in somy, which may very well explain the current data shown. In fact, the 3 parents [THY_4327, MCQ_1491, TAZ_4177] that did have aneuploid clones associated with them were generally disomic in the Supplemental Dataset and may support haplotype selection to explain their origin [see Barja et al, 2017. *Nature Ecol Evol*].

Great thanks to Reviewer 2 for the very thorough evaluation and precise suggestions for revision. As we mention in our initial statement, additional discussion around the mechanism of ‘non-Mendelian’ hybridization we observed in Cluster 2 rather detracted from our main finding of clear signatures of frequent meiotic sex we observed in Cluster 1. To shed further light of the mechanism of exchange in Cluster 2, we took the reviewer’s advice and tested whether *T. cruzi* karyotypes in this study may show instability during short-term laboratory handling or environmental changes. We confirmed this by resurrecting (after cryopreservation) and re-sequencing aneuploid Cluster 2 clones THY_4332_CL3, THY_4326_CL1 and TAZ_4174_CL4 at high read-depth (Tbl. S1b), then repeating somy analysis (Fig. S12). THY_4332_CL3 appeared to have lost several of its aneuploidies by the time of second sequencing (Fig. S12c). We also obtained multiple subclones (via limiting dilution cloning) from each of these three re-sequenced parent clones (Tbl. S1c). Somy levels in subclone TAZ_4174_CL4 ‘T2_D1’ were no longer consistent to the parent clone, providing additional indication of subcloning artefact and/or mosaic aneuploidy within the uni-clonal cell population (Fig. S12a). We have therefore cut unnecessary discussion of possible para-sexuality from the manuscript. This was never an essential hypothesis of the paper, only regrettably inflated in our first submission. However, we would like to note that non-cloned parent cultures such as MCQ_1491_MIX or TAZ_4177_MIX did not differ significantly from their derivative clones as the reviewer stated they did, i.e., that there had been direct evidence of plate-cloning artefact at the time of first submission.

Furthermore, 7 other parent populations were presented in Figure 2, but did not have clones associated with them {THY_4328, THY_3978, THY_4331, TBM_2798, TGM_4157, TAZ_4179, TAZ_3968}. What was the rationale for excluding parent populations from the population genomic analyses when the fineSTRUCTURE patterns did not suggest they were mixed infections.

Our rationale was in part due to aberrant FIS (most negative values in the dataset) in THY_4328_MIX, THY_3978_MIX, TAZ_4177_MIX, TAZ_4179_MIX, and TAZ_3968_MIX, as well as the increased presence of rare variants (sites with < 0.01 minor allele frequency) in THY_4328_MIX and MCQ_1491_MIX, latter of which later can be interpreted to harbor recombinant clones based on phylogenetic network relationships (Fig 1a). Given such observations we decided to take the definitive (albeit expensive) approach to exclude non-cloned strains in clone analysis even if they showed some uni-clonal features in fineSTRUCTURE analysis.

Interestingly, the two clones from MCQ_1491 were in fact genetic recombinants (see Fig 1a) and chromosome painting at core genome resolution of the parent against these two clones would likely provide important insight into the genetic mechanisms at play. For example, do blocks of heterozygosity exist in the parent, that are phased into discrete haplotypes in the “recombinant” clones?

We would urge caution in over-interpreting an individual splits-tree branch pair (Figure 1a) against the back-drop of widespread recombination in Cluster 1. Nonetheless, we can indicate that new chromosome painting analysis we undertook (e.g., Fig. S9) revealed that areas of dissimilarity between this particular sample-pair take place predominantly in heterozygous sequence patches, within which phasing analyses show that often neither haplotype A nor haplotype B is shared between the two clones. However, given a N=45 sample set that does not contain divergent progenitor genotypes, dissection of phased haplotype contributions down to the level of individual mutations or short sequence regions, as would be necessary for comparisons among closest relatives MCQ_1491_CL2 and MCQ_1491_CL3 and their non-cloned source, is not robust. For accurate results, we limited phasing analysis to assessment of pair-wise phylogenetic distances summed across larger, 60 kb sequence bins (Fig. S4). Nevertheless, as explained in response to Specific Comment 1 below (also Fig. 4, Panels 2-3, and Fig. S7) we identify plenty of recombination signals from homozygous regions in other pairs/sets of clones.

Specific Comments:

1 Both the Neighbour-Net and Fine Structure plots show evidence of genetic recombination and shared ancestry, but not their position or coverage within the core genome, to support the finding that these parasites undergo classical meiotic reproduction. Fine Structure suggests ~10 distinct ancestries across the two populations, how have these distinct ancestries been inherited genome-wide? Visualizing the extent of this, across the different strains examined would be a powerful figure to show the mosaic nature and the frequency of identifying different admixtures within the population genetics of this parasite. It is not currently clear whether admixture has occurred in the majority of strains - or just relatively few. Failure to show positionally across each genome the number of ancestries that exist, and whether/how they have recombined across the phased genomes is a deficit in the manuscript. Indeed, visualization tools exist that have been employed in a number of papers to facilitate depicting the types of relationships that are being advocated herein. See for example Rogers et al, PLoS Genetics, 2014 Fig 5, or Bajra et al, Nature Ecol Evol, 2017 Fig 4. Likewise, inclusion of a bottlebrush plot, if number of ancestries is limited, would be an informative pair-wise comparison that could be generated genome-wide, to identify the extent to which blocks of heterozygosity are consistent with a meiotic/genome-fusion process.

We have now implemented a number of genome-wide visualizations of ancestry variation that specify positions and frequency of genealogical shifts that indeed occur in a majority of Cluster 1 genomes. In the new section beginning at line 152, after network, fineSTRUCTURE and linkage decay analyses, Fig. 4 (Panel 1) now presents a genome-wide comparison of windowed alternate allele-frequency means (AAFMs) among clones. We further explicitly show that sequential 5 kb windows with similar AAFM values equate to large segmental increases in sequence-sharing among alternating sets of clones in Cluster 1 (Fig. 4, Panels 2-3, and Fig. S7). The fluctuation in pairwise sequence-similarity is also illustrated by SNP-painting chromosome and genome-wide (Figs. S8 and S9) following approaches in Barja et al. 2017 and Rogers et al. 2014 as suggested. We complement AAFM and SNP-painting plots with a new approach known as windowed topology weighting by iterative sampling of subtrees (“Twisst”) introduced by Martin and van Belleghem (Genetics 2017). This approach clarifies whether, where and how often genealogical relationships change within the genome. Basically, a subset of samples is chosen to denoise the complexity of the full phylogenetic tree. The empirical support for each possible topological arrangement for that sample subset is then calculated in sliding window analysis across the genome. We show how a single topology dominates genome-wide in Cluster 2 groups El Huayco and Ardanza (Fig. 4, Panel 4, and Fig. S10), whereas different topologies recurrently gain and lose support in the Bella Maria region (Fig. 4, Panel 3). The specific taxon topologies supported at each genomic position affirm

results from prior AAFM and SNP-painting plots. AAFM patch-sharing between numerous subsets of clones in Fig. 4, Panels 1 and 2, and visual inspection of SNP-painting in Fig. S9 for Bella Maria indicates that admixture occurs ubiquitously and that reticulate phylogenies are not driven by a single or few outlier samples. Multiple recombinant samples are required to generate the observed linkage decay and all fifteen possible 5-taxon Twisst topology permutations occur abundantly in the Bella Maria sample set (see Fig. R3, top plot, in Appendix of this document) as opposed to constant single-topology support in Cluster 2 (App. Fig. R3, middle and bottom plots).

2. In Figures 3c and 3d, the pairwise SNP association plots establish that the populations in EH and AR are not in linkage disequilibrium along chromosome 1, which is supported by the reticulate patterns identified in the Neighbour-Net plot in Fig 1a. The structuring in the pattern of SNP variant-associations is consistent with recombination of a limited number of alleles across the strains examined, underpinning why the chromosome painting analysis above should be performed. What is clear is that the SNP density is relatively low (see Figure 5b), and the patchy heterozygosity observed in the populations outside BM could be consistent with an asexual expansion model, so additional analyses proposed should be performed to better inform the exact phylogenomic relationship that exists between strains in this region.

The SNP-painting in Fig. S9 and complementary AAFM / Twisst analyses have helped affirm the scarcity of recombinant loci in Cluster 2. Occasional shifts from heterozygous to common homozygous genotypes do not statistically relate to coding vs. non-coding sequence annotation in Cluster 2 ($\chi^2 = 0.089$, $df = 1$, p -value = 0.764, noted in Fig. S9 legend), suggesting rather that rare introgression events may underlie subtle instances of haplotype dis-linkage observed in these groups – e.g., in Gerinoma and TCQ clones (lines 173 – 179 and 231 – 237). SNP density variation in Cluster 2 (App. Fig. R4) is rather noisy, likely because the genome was heavily masked, both in large continuous intervals and in smaller intermittent snips throughout. Nonetheless, heterozygosity is generally not divided into discrete patches to the extent observed in Cluster 1 (Fig. 4, Panel 1). In the face of predominant non-private heterozygosity across Cluster 2 – also large stretches of phased heterozygous genotype-sharing, nuclear phylogenetic reticulation and mitochondrial genetic connectivity among these clones – we do not see how the Meselson Effect should find any support from insubstantial shifts in SNP density throughout the genome.

3. Figure 6 is quite difficult to interpret. It would appear that MBM_1466 shares regions of the same haplotype as TRT_3949, but differs in heterozygosity in small blocks on chromosome 5. Could this be the result of random mutation consistent with genetic drift? And that TAZ_4173 and THY_4334 likewise share similar haplotypes, but differ in the extent of their heterozygosity, but are conserved relative to others shown. It is therefore important to see this analysis done genome-wide. Is it possible to perform this using concentric tracks, depicting heterozygosity in one track, and chromosome painting on the other?

We have removed Fig. 6b because the information was misleading and now redundant to the new AAFM plot (Fig. 4). The non-drift like ancestry between multiple Cluster 1 clones now manifests clearly as episodic sharing of multiple consecutive windowed AAFM values and in the shape of fluctuating topology weighting functions, that is, topology curves that clearly dominate over others in discrete sequence intervals, running not simply as a series of narrow spikes (i.e., noise) across the genome. Please see also cumulative topology support values in App. Fig. R3. All topologies gather substantial support over the course of genome-wide 50 kb windowed analysis, in contrast to those from Cluster 2. Regarding concentric alignments, we decided to plot various allele-frequency binning, SNP-painting and phylogenetic topological comparisons in horizontal format; we found this better suited for our sample size.

4. This entire analysis would benefit from showing a genome-wide SNP density plot, to formally show that the SNP density is low, and to more readily highlight regions of increased SNP density that likely represent introgression from distinct genetic ancestries that have undergone an independent evolution. This is particularly relevant for the strains outside of BM, as this analysis may help to shed light on whether the relationship among strains is one of mutational drift by asexual expansion, or introgression of a limited number of haplotypes followed by haplotype selection/gene conversion to a homozygous state.

As noted in response to related query in Specific Comment 2, Fig. S9 summarizes all SNPs together with coding vs. noncoding sequence annotation, though these functional annotations do not relate to occasional regions of increased homozygous SNP-sharing observed in Cluster 2 (p -value = 0.764). In other words, occasional genomic areas of common homozygous mutation among Cluster 2 samples do not support an independent selection effect. Cluster 2 genomes also generally lack substantial loss-of-heterozygosity tracts (recall Tbl. S2) as might have been expected under aneuploid haplotype selection or canonical gene conversion events. Rather, geographically separate El Huayco and Ardanza share a majority of their heterozygous sites and clone pairs occasionally share identity on both haplotypes in phased heterozygous tracts (Fig. S4). Nevertheless, we attached a SNP density plot for a subset of best-mapping chromosomes for your interest (App. Fig. R4). We find it unsuitable to include in the manuscript, given that

extensive reference-masking creates sequence bins with few of no SNP sites, and unmasked coding-regions are biased against those most likely to experience diversifying selection (see response to Specific Comment 5). In App. Fig. R4, however, the key point is that density graphs for SNP sites in El Huayco are extremely similar to those for Ardanza, consistent with ancient hybridization as a source of genome-wide heterozygosity in contrast to pure asexual expansion leading to SNP variation private to individuals within the two groups

5. Is it known whether *T. cruzi* is prone to mitotic recombination, in similar to what is seen in the var genes for *P. falciparum*? The pair-wise SNP density plots, overlaid with gene model information may be informative to determine if the biallelic hot-spots may represent gene family expansions, which could produce regions of patchy heterozygosity that appear less-random.

Indeed, mitotic recombination has been emphasized in creating antigenic diversity in *T. cruzi* (Talavera-Lopez et al. 2018, biorxiv) but heterozygosity will spike artefactually in an unmasked *T. cruzi* genome. Unfortunately, the masked genome filters heavily against repetitive areas of surface gene expansion – which make up the vast majority of multi-gene families in *T. cruzi*. In the masked genome, patches of high AAFM in Cluster 1 genomes (Fig. 4) are actually more common in non-coding sequence regions ($\chi^2 = 173.12$, $df = 1$, $p\text{-value} < 0.001$), arguing against a convergent selective effect, especially for unfixed homozygous AAFM patches, i.e., those shared between few clones. However, we largely refrain from interpretation of such statistics because masked analyses will bias coding areas towards single-copy/non-expanding genes.

6. Fig S11, No outgroup for A is depicted – the figure would be more meaningful if additional reference strains were added, including TcI Sylvio, TcII Esmeraldo, TcVI Brener? In Fig S11b, CytC – the analysis is incomplete. Where do the other strains analysed from Ecuador cluster – with Tc1, to distinguish just how different 3087 is?

For clarification we have now included the other strain genotypes and those of representative from all *T. cruzi* DTUs (including TcBat) as well *T. c. marinkellei* and *T. dionisii* in the cytochrome B plot. This provides greater resolution than few outgroups available for the full maxicircle analysis.

Appendix

Tbl. R1 Population genetic descriptive metrics for *T. cruzi* I clones from Bella Maria (BM; associated to Cluster 1), El Huayco (EH; Cluster 2) and Ardanza (AR; Cluster 2). Abbreviations: Poly. (Polymorphic); MAF (within-group Minor Allele Frequency); HW (Hardy-Weinberg); Het. (Heterozygous).

a) Analysis using only 1 clone per vector/host.

Group (n)	Poly. Sites	Median Nucleotide Diversity (π) per site	Median Watterson Estimator (θ) per site	% Poly. Sites at MAF > 0.05	Private Sites (vs. BM / EH / AR)	Singleton Sites	Poly. Sites in HW Equilibrium	Het. Sites	Fixed Het. Sites
BM (15)	95,313	0.13	0.001	59 %	0 / 41,270 / 41,063	22,344	90,461	55,571	2,848
EH (12)	76,889	0.53	0.001	71 %	22,846 / 0 / 17,681	6,855	44,911	56,016	45,792
AR (11)	75,709	0.55	0.001	71 %	21,459 / 16,501 / 0	9,844	72,968	55,638	47,761

b) Analysis using ≥ 1 clone per vector/host (original results).

Group (n)	Poly. Sites	Median Nucleotide Diversity (π) per site	Median Watterson Estimator (θ) per site	% Poly. Sites at MAF > 0.05	Private Sites (vs. BM / EH / AR)	Singleton Sites	Poly. Sites in HW Equilibrium	Het. Sites	Fixed Het. Sites
BM (15)	96,691	0.09	0.001	48 %	0 / 40,177 / 40,262	14,013	87,500	58,102	2,134
EH (12)	80,052	0.15	0.001	70 %	23,538 / 0 / 18,016	4,525	33,980	58,980	44,945
AR (11)	78,325	0.16	0.001	71 %	21,896 / 16,289 / 0	6,064	35,799	58,392	45,287

Fig. R1 Results for simulated datasets without recombination (BS_n: $\rho = 0$; see *Methods*) show that sample sizes ≤ 6 fail to reject the null-hypothesis of Hardy Weinberg Equilibrium (HWE) at a majority of (100% independently occurring) polymorphic sites.

Linkage with distance, chr1 (Bella Maria, only 1 clone per host/vector)

Linkage with distance, chr1 (El Huayco, only 1 clone per host/vector)

Linkage with distance, chr1 (Ardanza, only 1 clone per host/vector)

Fig. R2 Linkage decay in *T. cruzi* I groups with only one clone per vector/host used in analysis. The top panel shows decay of linkage disequilibrium on chromosome 1 for *T. cruzi* I clones from Bella Maria. Average pairwise linkage values (r^2) among all diagnostic SNP sites are plotted for map distance classes between 0 and 100 kb. The second and third panels show lack of linkage decay on chromosome 1 for *T. cruzi* I clones from El Huayco (middle) and Ardanza (bottom).

Fig. R3 Given five *T. cruzi* I clones that span the phylogenetic clade topology of the Bella Maria sample set (Cluster 1), all 15 possible subtree topologies accumulate abundant support from different regions of the genome (green circles in top plot). By contrast, a single topology dominates for 5-sample subsets representative of phylogenetic diversity in El Huayco (grey circles, middle plot) and Ardanza (blue circles, bottom plot) groups of Cluster 2. Please notice much larger y-axis maxima in middle and bottom plots.

Fig. R4 SNP density on chromosomes 1, 5, 18, 19, 21 and 26 in *T. cruzi* I sample genomes from Cluster 2. SNP counts (y-axis) are plotted for sequential 10 kb sequence windows across the masked genome. Each chromosome (except chromosomes 17, 40 and 47) is represented by a separate plot. Grey curves show SNP density in El Huayco genomes. Blue curves represent the Ardanza sample set.

Reviewers' comments:

Reviewer #1 (Remarks to the Author):

The authors have submitted a revised MS alongside their responses to the reviewers' comments. While the authors have clarified some points, they have not satisfactorily addressed all the issues raised and the MS still lacks clarity in places.

For example, the numbers of clones in lines 90-95 still add up to 43 not 45. While the numbers of clones are explained in detail in the authors' response letter, this will be of no help to the reader of the final published paper. Similarly, the authors claim to have addressed the confusion in the Fig 2 legend in their response letter, but the legend is unchanged: it still states 3 samples with horizontal banding whereas the figure clearly shows 4. Sadly, there are several other examples like this where the authors have explained what they mean in their response letter, but have not actually clarified this in the MS by substantively altering the text (e.g. "eclectic but stercorarian" remains, even to the extent of not correcting the spelling mistake in stercorarian). I can only conclude that the authors have not carried out a very thorough revision of the MS.

Reviewer #2 (Remarks to the Author):

Refocusing the manuscript on the frequency and distribution of admixture and by incorporating quantitative phylogenetic analyses and chromosome painting to better illustrate the concepts they advance in their population genetic datasets has vastly improved their manuscript. This is important work, both for its perspective, and for its ability to promote new research into the various replication modes utilized by *T. cruzi* to promote its transmission in nature.

My only hesitation is in the use of the word "Extreme heterozygosity" used in various places (lines 262, 265, 280, 282, for example). "Extant heterozygosity" may be more appropriate. When two distinct strains hybridize that are largely homozygous, they become heterozygous at every variant position between the two strains, and these "offspring" represent full genomic hybrids - should this be referred to as "extreme heterozygosity"? Extreme is such a strong adjective. Although I appreciate this is terminology adopted from others, however, the authors have previously cautioned on adopting generalized theories, such as PCE, so perhaps its best to stay focused on the truth of their data, which does not show extreme heterozygosity. This is a term often applied to strains that are evolving largely asexually that have accumulated significant numbers of private SNPs, that altogether (across many genomes sampled) give an impression of extreme heterozygosity.

Minor revision:

Line 317 - please replace "stercocarian" with stercorarian" to correct the spelling

We thank both reviewers for their extremely valuable time and expertise spent in evaluation of our manuscript.

Reviewer 1 stated: *The authors have submitted a revised MS alongside their responses to the reviewers' comments. While the authors have clarified some points, they have not satisfactorily addressed all the issues raised and the MS still lacks clarity in places.*

We have inspected the manuscript for any pieces of text in which immediate clarity could be further enhanced. We have introduced amendments to lines 62-64, 90-94, 98-104, 115, 144-145, 191, 195, 216, 225-226, 246-247, 253, 260, 263, 277-281, 309, 312-315, 350, 354-356, 402, 461-462, 769-772 and 830.

Reviewer 1 stated: *For example, the numbers of clones in lines 90-95 still add up to 43 not 45. While the numbers of clones are explained in detail in the authors' response letter, this will be of no help to the reader of the final published paper.*

We hope that amended lines 90-94 now clear up any confusion about the $n = 45$ sample sum. Before we had not explicitly referred to TCQ and TRT 'outlier' clones at this stage in the text (not until Tbl. S1a and Fig. 2 (see legend), and then in more detail in the last subsection of the Results). These outliers are now clearly highlighted in lines 91-92 and 94, respectively. Also, we now make earlier reference to Tbl. S1a (line 90), where all clones and sampling locations are detailed.

Similarly, the authors claim to have addressed the confusion in the Fig 2 legend in their response letter, but the legend is unchanged: it still states 3 samples with horizontal banding whereas the figure clearly shows 4.

We have now updated the Fig. 2 legend to state the correct number of bands (line 769).

(e.g. "eclectic but stercorarian" remains, even to the extent of not correcting the spelling mistake in stercorarian).

We have fixed the spelling (line 313). The context around "stercorarian" has also been further amended for clarity. We regret these mistakes survived until now if they did really give the impression of an un-thorough revision, because in fact we devoted major effort (several months) in re-analysis and re-write.

A final point of unresolved clarity with regards to our previous response to Reviewer 1 relates to their comment:

*Overall, the number of individual hosts sampled from each area is quite small – because a number of clones were examined from each host, this has increased the sample size. However, clones from the same host are not independent samples – indeed they often seem to be genetically related – so this is not a random sample from the *T. cruzi* population in this area. How does this factor into the population genetics stats?*

We had responded (but only to the reviewer) that we believe there is no single best model data set / sampling configuration that meaningfully describes population genetic structure in a facultative clonal/sexual microorganism that occurs both in homogeneous and multiclonal infections distributed across different yet interacting vectors and hosts. We then provided the reviewer with re-calculations of all population genetic metrics in Tbl. 1 and repeated linkage decay analyses with only one representative clone per infection. Inference did not change. We now clearly mention these findings in the section on linkage decay (lines 143-145), with figures available upon request, and also notify about these validations in the Methods section (461-462).

We believe we have now fully satisfied the Reviewer 1's concerns.

Reviewer 2, thank you for your noting this last concern about the term “extreme heterozygosity”. We had not taken into account its connotations and now changed the wording in all cases, including also in the discussion of Weir et al. 2016 as not to generalize or draw unwanted extra interpretation (lines 260, 263, 277, 278 and 280-281). Of course, also the ‘stercorarian’ mistake (line 313) has also been corrected.

Thanks for your renewed consideration of this work.